# UNIQ: Offline Inverse Q-learning for Avoiding Undesirable Demonstrations

## Abstract

We address the problem of offline learning a policy that avoids undesirable demonstrations. Unlike conventional offline imitation learning approaches that aim to imitate expert or near-optimal demonstrations, our setting involves avoiding undesirable behavior (specified using undesirable demonstrations). To tackle this problem, unlike standard imitation learning where the aim is to minimize the distance between learning policy and expert demonstrations, we formulate the learning task as maximizing a statistical distance, in the space of state-action stationary distributions, between the learning policy and the undesirable policy. This significantly different approach results in a novel training objective that necessitates a new algorithm to address it. Our algorithm, UNIQ, tackles these challenges by building on the inverse Q-learning framework, framing the learning problem as a cooperative (non-adversarial) task. We then demonstrate how to efficiently leverage unlabeled data for practical training. Our method is evaluated on standard benchmark environments, where it consistently outperforms state-of-the-art baselines.

## 1 Introduction

Reinforcement learning (RL) is a powerful framework for learning to maximize expected returns and has achieved remarkable success across various domains. However, applying reinforcement learning to real-world problems is challenging due to difficulties in designing reward functions and the requirement for extensive online interactions with the environment. While some approaches have addressed these challenges, they often rely on costly datasets, requiring either accurate labeling or clean, consistent data, which is often impractical. Imitation learning (Abbeel & Ng, 2004; Ziebart et al., 2008; Kelly et al., 2019) offers a more feasible alternative, enabling agents to learn directly from expert demonstrations without the need for explicit reward signals. It has proven effective in several tasks, even with limited expert data, and is particularly useful in capturing human preferences.

Most existing imitation learning approaches prioritize maximizing task performance (i.e., expected return) by closely mimicking expert demonstrations (Ho & Ermon, 2016; Fu et al., 2018; Kostrikov et al., 2019; Garg et al., 2021). However, in practice, expert or near-expert demonstrations may be unavailable or insufficient. In many scenarios, instead of high-quality examples, there may be collections of undesirable (or suboptimal) demonstrations that should be avoided. For example, in a large dataset of user conversations used to train a chatbot, the system must learn to avoid inappropriate or sensitive content that is present in the data (Duan et al., 2024). Similarly, in the development of self-driving cars, while companies may collect user driving data to train their models, the system must ensure it does not replicate faulty behavior, such as traffic violations or unsafe driving practices (Bansal et al., 2018). In the field of treatment optimization, data may include actions that led to bad patient outcomes and the system must learn to avoid such behaviors. In these cases, avoiding undesirable behaviors is essential to training a safe and effective model. These types of undesirable demonstrations are common in real-world applications and are crucial for shaping the desired policy.

Although this is an important and interesting problem setting, there has been limited research addressing it effectively. Here, we formulate this challenge as an *Offline Reverse Imitation Learning problem*: given a dataset containing *undesired demonstrations* we wish to avoid, alongside a much larger unlabelled dataset consisting of both desired and undesired demonstrations, the goal is to learn

a desired policy. The key question is whether we can leverage the undesired demonstrations to learn a policy that avoids undesirable behaviors.

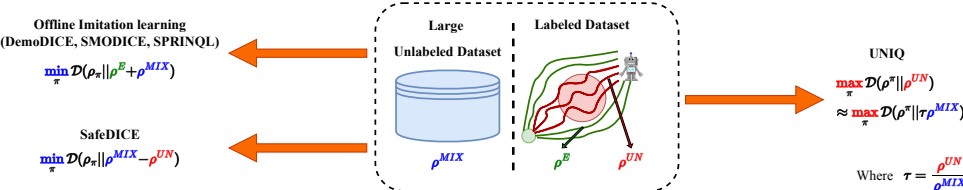

Figure 1: A general overview of offline imitation learning approaches: while prior approaches mostly aim at minimizing the distance between the learning policy and expert data (or a combination of non-expert and unlabeled data), our goal is to maximize the distance between the learning policy and an undesired policy. *In UNIQ, the unlabeled dataset ($\rho^{\mathrm{Mix}}$) is used in the empirical approximation of the main objective function.*

To the best of our knowledge, only SafeDICE (Jang et al., 2024) directly addresses this setting by mixing the undesirable and unlabeled datasets, assigning negative weights to the former, and then mimicking the mixed policy. This approach, however, may suffer from the fact that the mixed policy is not necessarily a desirable one to follow, which is often the case in practice. There are existing methods that, while not specifically designed to address this problem, can be adapted to handle it. For instance, preference-based RL methods (Brown et al., 2019; Lee et al., 2021b; Hejna & Sadigh, 2024) can be modified by creating pairwise comparisons between the undesirable and unlabeled demonstrations, assuming the unlabeled demonstrations are more preferred. However, this approach is indirect and may not be effective. Similar to SafeDICE, this approach will also suffer as the unlabeled dataset may contain undesirable demonstrations that are not necessarily preferable to those in the undesirable dataset. Another approach, Discriminator-Weighted Behavioral Cloning (DWBC) (Xu et al., 2022), can be adapted to train a discriminator to distinguish between desirable and undesirable demonstrations, allowing the agent to avoid learning from undesirable samples. Our experiments later will show that our method significantly outperforms the above baselines on benchmark problems.

In this paper, we develop a principled framework for learning from undesirable demonstrations, based on the well-known MaxEnt RL framework (Ziebart et al., 2008) and inverse Q-learning (Garg et al., 2021) — a state-of-the-art imitation learning method. In our work, unlike traditional imitation learning approaches where the main goal is to minimize the distance between the learning policy and expert (or near-optimal) demonstrations, we instead aim to maximize a statistical distance between the learning policy and the undesirable policy (represented by undesirable demonstrations) in the state-action stationary distribution space. To address this novel learning objective, we adopt the inverse Q-learning framework. In our context, since this objective requires optimization in the opposite direction as compared to the standard IQ-learn method (Garg et al., 2021), it demands significantly novel investigations. Specifically, our contributions can be summarized as follows:

- We demonstrate that maximizing the statistical distance between the learning policy and the undesirable policy can be formulated as a cooperative training task, in contrast to previous imitation learning approaches where the objective is adversarial Ho & Ermon (2016); Fu et al. (2018); Kostrikov et al. (2019). This new cooperative training objective poses new challenges that existing algorithms cannot readily solve. To tackle this, we show that certain beneficial characteristics of the standard IQ-learn algorithm can be brought over in our context – the training problem can be equivalently reformulated as a minimization problem over the Q-space, where the optimal policy is recovered as a soft-max of the Q-function.

- To efficiently solve the training problem in the Q-space using limited undesirable demonstrations, we introduce an occupancy correction to recast the training objective so that the expectation over the undesirable policy can be empirically approximated by unlabeled trajectories. This approach offers three main advantages: (i) the unlabeled data contains many more samples than the undesired data set, so using unlabeled demonstrations would greatly improve the accuracy of the empirical approximations, (ii) the training outcome is theoretically independent of the quality of the unknown random policy (represented by the

unlabeled data), and (iii) it does not require an additional hyper-parameter since we do not combine the two datasets (undesired and unlabeled) as in most prior approaches. Moreover, we show that the occupancy correction can be estimated by solving a convex optimization problem using samples from both undesirable and unlabeled data.

- For policy extraction, we employ a weighted behavior cloning (WBC) formulation. This WBC approach theoretically ensures to recover the optimal policy derived from the Q-learning process while providing more stable offline training outcomes, compared to directly extracting the policy from the Q-function. While our algorithm, UNIQ, tackles a significantly different and more challenging setting compared to the original IQ-learn, it only requires a *minimal adaptation* from the IQ-learn algorithm, with two additional steps — computing the occupancy correction and extracting the policy via WBC — which can be carried out quickly and efficiently.

- We evaluate our method on the Safety-Gym and Mujoco-velocity benchmarks (Ray et al., 2019; Ji et al., 2023), two of the most challenging environments in constrained reinforcement learning, and demonstrate superior performance compared to several state-of-the-art baselines.

## 2 RELATED WORK

**Imitation learning.** Imitation learning is a key technique for learning from demonstrations. Behavioral Cloning (BC) maximizes the likelihood of expert demonstrations but often struggles due to distributional shift (Ross et al., 2011). To improve, Generative Adversarial Imitation Learning (Ho & Ermon, 2016; Fu et al., 2018) aligns the learner's policy with the expert's using GANs (Goodfellow et al., 2014), while SQIL (Reddy et al., 2019) assigns simple rewards to expert and non-expert demonstrations to learn a value function. PWIL (Dadashi et al., 2021) uses the Wasserstein distance (Vaserstein, 1969) to compute rewards. While these methods show promise, they rely on online interaction, which can be impractical. For offline learning, AlgaeDICE (Nachum et al., 2019) and ValueDICE (Kostrikov et al., 2020) use Stationary Distribution Correction Estimation (DICE) but face stability issues. Inspired by ValueDICE, O-NAIL (Arenz & Neumann, 2020) introduced an offline method without adversarial training. IQ-learn (Garg et al., 2021), a popular approach, supports both online and offline learning and offers a state-of-the-art framework with several variants developed based on it (Al-Hafez et al., 2023; Hoang et al., 2024b). Unlike the above mentioned works, our focus in this paper is on offline **reverse** imitation learning that aims to avoid undesirable trajectories, as opposed to imitating expert trajectories. As indicated earlier, this requires fundamentally different methods due to the nature of the problem.

**Imitation Learning from Sub-optimal Demonstrations.** There are two main research directions in this area. The first focuses on online and offline preference-based imitation learning methods. Online approaches, such as T-REX (Brown et al., 2019), PrefPPO (Lee et al., 2021a), and PEBBLE (Lee et al., 2021b), leverage ranked sub-optimal demonstrations to learn a preference-based reward function using the Bradley-Terry model (Bradley & Terry, 1952). While these methods achieve strong performance, they rely on interaction with the environment. In contrast, offline methods, such as those proposed by (Kim et al., 2023; Kang et al., 2023; Hejna & Sadigh, 2024), rely heavily on extensive pairwise trajectory comparisons. SPRINQL (Hoang et al., 2024b) addressed this limitation by utilizing demonstrations categorized into different levels of expertise, resulting in better performance with fewer comparison demands. While these methods concentrate on imitating preferred (or expert) trajectories, our focus is on avoiding non-preferred (or undesired) trajectories. This important distinction necessitates significant changes in methodology.

The second direction focuses the use of additional unlabeled datasets to enhance learning from expert data. Beginning with DemoDICE (Kim et al., 2021), several DICE-based methods (Ma et al., 2022; Kim et al., 2022; Yu et al., 2023) have been developed to utilize small sets of expert demonstrations, supplemented by larger unlabeled datasets. In addition to these DICE-based methods, DWBC (Xu et al., 2022) propose a simple and efficient method based on training a classifier using positive-unlabeled learning (Kiryo et al., 2017). SafeDICE (Jang et al., 2024) presents a DICE-based framework capable of learning from undesirable demonstrations. This method combines an undesirable policy (represented by an undesirable dataset) with a random policy (represented by a larger unlabeled dataset), assigns negative weights to the undesirable policy, and then applies a

standard DICE-based approach (Kim et al., 2021; 2022) to mimic the combined policy. In our paper, we advance this line of work by developing an new inverse Q-learning algorithm (Garg et al., 2021) where the primary goal is to maximize the statistical distance between the learning policy and the undesirable policy. Our method offers several advantages over prior approaches, including minimal hyper-parameter tuning and reduced sensitivity to the quality of the unlabeled data.

## 3 BACKGROUND

**Preliminaries.** We consider a MDP defined by the following tuple $\mathcal{M} = \langle S, A, r, P, \gamma, s_0 \rangle$, where $S$ denotes the set of states, $s_0$ represents the initial state set, $A$ is the set of actions, $r : S \times A \to \mathbb{R}$ defines the reward function for each state-action pair, and $P : S \times A \to S$ is the transition function, i.e., $P(s'|s, a)$ is the probability of reaching state $s' \in S$ when action $a \in A$ is made at state $s \in S$, and $\gamma$ is the discount factor. In reinforcement learning (RL), the aim is to find a policy that maximizes the expected long-term accumulated reward $\max_\pi \left\{ \mathbb{E}_{(s,a) \sim \rho_\pi}[r(s, a)] \right\}$, where $\rho_\pi$ is the occupancy measure of policy $\pi$: $\rho_\pi(s, a) = (1 - \gamma)\pi(a|s)\sum_{t=1}^\infty \gamma^t P(s_t = s|\pi)$.

**MaxEnt IRL** The goal of MaxEnt IRL is to derive a reward function $r(s, a)$ based on a set of expert demonstrations, $\mathcal{D}^E$. Let $\rho^E$ denote the occupancy measure of the expert policy. The MaxEnt IRL framework, as introduced by (Ziebart et al., 2008), aims to recover the expert's reward function by optimizing the following expression: $\max_r \min_\pi \left\{ \mathbb{E}_{\rho^E}[r(s, a)] - \mathbb{E}_{\rho_\pi}[r(s, a)] - H(\pi) - \psi(r) \right\}$, where $H(\pi) = \mathbb{E}_{\rho^\pi}[-\log \pi(s, a)]$ is the discounted causal entropy of the policy $\pi$ and $\psi(r) : \mathbb{R}^{S \times A} \to \mathbb{R}$ is a convex reward regularizer. In essence, the objective is to identify a reward function that maximizes the gap between the expected reward under the expert's policy and the maximum expected reward across all other policies (as determined by the inner minimization).

**Inverse Q-learning (IQ-Learn) from expert demonstrations.** Given a reward function $r$ and a policy $\pi$, the soft Bellman equation is defined as $\mathcal{B}_r^\pi[Q](s, a) = r(s, a) + \gamma\mathbb{E}_{s'}[V^\pi(s')]$, where $V^\pi(s) = \mathbb{E}_{a \sim \pi(a|s)}[Q(s, a) - \log \pi(a|s)]$. The Bellman equation $\mathcal{B}_r^\pi[Q] = Q$ is contractive and always yields a unique Q solution (Garg et al., 2021). In IQ-Learn, they further define an inverse soft-Q Bellman operator $\mathcal{T}^\pi[Q] = Q(s, a) - \gamma\mathbb{E}_{s'}[V^\pi(s')]$. (Garg et al., 2021) show that for any reward function $r(a, s)$, there is a unique $Q^*$ function such that $\mathcal{B}_r^\pi[Q^*] = Q^*$, and for a $Q^*$ function in the $Q$-space, there is a unique reward function $r$ such that $r = \mathcal{T}^\pi[Q^*]$. This result suggests that one can safely transform the objective function of the *MaxEnt IRL* from $r$-space to the Q-space as follows:

$$\max_Q \min_\pi \ \Phi(\pi, Q) = \mathbb{E}_{\rho^E}[\mathcal{T}^\pi[Q](s, a))] - \mathbb{E}_{\rho_\pi}[\mathcal{T}^\pi[Q](s, a)] - H(\pi) - \psi(\mathcal{T}^\pi[Q](s, a))) \quad (1)$$

which has several advantages, namely, the objective function $\phi(\pi, Q)$ is concave in $\pi$ and convex in Q. Moreover, the inner problem $\min_\pi \phi(\pi, Q)$ has a closed form solution as $\pi^*(a|s) = \exp(Q(s, a))/\sum_a \exp(Q(s, a'))$. As a result, the maximin problem can be converted to a *non-adversarial* problem in the Q-space as:

$$\max_Q \ \mathbb{E}_{\rho^E}[\mathcal{T}[Q](s, a))] - (1 - \gamma)\mathbb{E}_{s_0}[V^Q(s)] - \psi(\mathcal{T}[Q](s, a))) \quad (2)$$

where $\mathcal{T}[Q](s, a)) = Q(s, a) - \gamma\mathbb{E}_{s' \sim P(s'|s, a)}[V^Q(s')]$ and $V^Q(s) = \log\left(\sum_a \exp(Q(s, a))\right)$, which is a softmax of the Q function. The reward function can then be recovered as $r^Q(s, a) = \mathcal{T}[Q](s, a)$. Thus, in 2, the objective can be interpreted as training a reward function (via a Q-function) that maximizes the expected reward under the expert policy while minimizing the overall expected reward.

## 4 UNIQ: INVERSE Q-LEARNING FROM UNDESIRED DEMONSTRATIONS

We now introduce UNIQ, our framework for inverting a Q function based on undesired demonstrations. This approach can be seen as a *reverse version* of the standard Inverse Q-learning algorithm (Garg et al., 2021), where the goal is *not to imitate* but rather to *avoid undesired behaviors.*

## 4.1 Inverse Q-learning from Undesired Demonstrations

In our setting, we have a set of undesired demonstrations, denoted as $\mathcal{D}^{\text{UN}}$, along with a supplementary set of unlabeled demonstrations, denoted as $\mathcal{D}^{\text{MIX}}$. The unlabeled dataset $\mathcal{D}^{\text{MIX}}$ may contain a mix of random, undesired, and expert demonstrations, and it will be used to support offline learning. Let $\rho^{\text{UN}}$ be the occupancy measure (or stationary distribution) of the undesired policy (represented by the undesired dataset). Adapting the MaxEnt RL framework, we consider the following learning objective:

$$\min_{\pi} \min_{r} \left\{ L(\pi, r) = \mathbb{E}_{\rho^{\text{UN}}}[r(s, a)] - \mathbb{E}_{\rho_\pi}[r(s, a)] - H(\pi) + \psi(r) \right\} \tag{3}$$

The objective can be interpreted as finding a reward function that assigns low rewards to the undesired demonstrations and high rewards to others, while also identifying a policy function that maximizes the expected long-term reward. It is important to note that, in our context, where the objective contrasts with the standard learning-from-demonstration scheme, the learning problem is no longer *adversarial* as in prior imitation learning approaches (Ho & Ermon, 2016; Kostrikov et al., 2019). Instead, it can be framed as a *cooperative learning problem*, where the objective is to jointly identify a policy and reward function that minimize the objective function $L(\pi, r)$.

To gain a deeper understanding of the objective function in Eq. 3, the following proposition demonstrates that solving Eq. 3 is indeed equivalent to **maximizing the statistical distance**, parameterized by $\psi$, between the undesired policy and the learning policy.

**Proposition 4.1.** *For a non-restricted feasible set of the reward function:*

$$\min_{r} \min_{\pi} \left\{ L(\pi, r) \right\} = - \max_{\pi} \{ d_\psi(\rho^\pi, \rho^{\text{UN}}) - H(\pi) \} \tag{4}$$

*where $d_\psi(\rho^\pi, \rho^{\text{UN}}) = \psi^*(\rho^\pi - \rho^{\text{UN}})$, and $\psi^*$ is the convex conjugate of the convex function $\psi$, i.e., $\psi^*(t) = \sup_z \{ \langle t, z \rangle - \psi(z) \}$.*

It can be observed that solving 3 directly encourages the learning policy to deviate as much as possible from the undesired policy, which is derived from undesirable demonstrations. This approach contrasts with that of SafeDICE (Jang et al., 2024), which minimizes the KL divergence between the learning policy and the mixed policy, enabling the use of standard imitation learning methods. The primary limitation of this approach is that, with the quality of the unlabeled dataset being unknown, imitating the mixed policy may not lead to the desired learning outcome. In contrast, our approach does not rely on such a combination. Instead, we focus on maximizing the statistical gap in 4, leveraging the unlabeled dataset to support the practical training of our primary objective in 3.

Even though the minimization problem Eq. 3 can be directly solved using standard optimization algorithms to recover a reward and policy function, prior research indicates that transforming Eq. 3 into the Q-space can improve efficiency. As discussed in Section 3, there is a one-to-one mapping between any reward function $r$ and a corresponding function $Q$ in the Q-space. Therefore, the minimization problem in Eq. 3 can equivalently be transformed as:

$$\min_{Q} \min_{\pi} \left\{ L(\pi, Q) = \mathbb{E}_{\rho^{\text{UN}}}[\mathcal{T}^\pi[Q](s, a)] - \mathbb{E}_{\rho_\pi}[\mathcal{T}^\pi[Q](s, a)] - H(\pi) + \psi(\mathcal{T}^\pi[Q](s, a)) \right\} \tag{5}$$

where $\mathcal{T}^\pi[Q](s, a)) = Q(s, a) - \gamma \mathbb{E}_{s'}[V^\pi(s')]$ and $V^\pi(s) = \mathbb{E}_{a \sim \pi(a|s)}[Q(s, a) - \log \pi(a|s)]$

Compared to the primary objective of the standard IQ-learn algorithm in Eq. 1, our objective function in Eq. 5 is no longer adversarial with respect to $Q$ and $\pi$. As a result, *there are questions regarding whether the key advantages of the IQ-learn algorithm—such as the closed-form for the optimization over $\pi$ and the concavity of the objective in the Q-space— would still hold with the new objective.* To address this, we introduce the following proposition, which states that if the regularizer function $\psi(\cdot)$ is non-decreasing, then the objective function $L(\pi, Q)$ is convex in $\pi$. Furthermore, the minimization problem $\min_\pi L(\pi, Q)$ retains a closed-form solution, thereby simplifying the learning objective.

**Proposition 4.2.** *The following statements hold:*

    *(i) The function $L(\pi, Q)$ is convex in $\pi$ and the problem $\min_\pi L(\pi, Q)$ has a unique optimal solution at $\pi^Q(s, a) = \frac{\exp(Q(s, a))}{\sum_{a'} \exp(Q(s, a'))}$.*

    *(ii) The learning objective function can be simplified as:*

$$\min_{Q} \min_{\pi} \left\{ L(\pi, Q) \right\} = \min_{Q} \left\{ \mathcal{F}(Q) = \mathbb{E}_{\rho^{\text{UN}}}[r^Q(s, a)] - (1 - \gamma) \mathbb{E}_{s_0}[V^Q(s_0)] + \psi(r^Q) \right\}$$

$$\text{where } V^Q(s) = \log\left(\sum_a \exp(Q(s,a))\right) \text{ and } r^Q(s,a) = Q(s,a) - \gamma\mathbb{E}_{s'\sim P(\cdot|s,a)}V^Q(s').$$

Proposition 4.2 shows that, similar to IQ-learn, our training objective can be framed as an optimization problem over the Q-space, where the optimal policy can be computed as the soft-max of the Q-function. However, there is a key difference lying in the nature of the objective: while in IQ-learn, the training objective is convex within the Q-space, this is not the case in our context, where the objective $\mathcal{F}(Q)$ is neither convex nor concave in $Q$.

We now discuss the learning within the Q-space:

$$\min_Q \left\{ \mathcal{F}(Q) = \mathbb{E}_{\rho^{\text{UN}}}[r^Q(s,a)] - (1-\gamma)\mathbb{E}_{s_0}[V^Q(s_0)] + \psi(r^Q) \right\} \tag{6}$$

The learning objective can be interpreted as finding a reward function (expressed as a function of Q) that assigns the lowest possible rewards to the undesired demonstrations while maximizing the overall expected rewards, $\mathbb{E}_{s_0}[V^Q(S_0)]$.

## 4.2 LEARNING WITH UNLABELED DATA

To solve Eq. 6, the expectation $\mathbb{E}_{\rho^{\text{UN}}}[r^Q(s,a)]$ can be empirically approximated using samples from the set of undesired demonstrations $\mathcal{D}^{\text{UN}}$. However, in our setting, this set is limited. Additionally, since the learning process must be conducted offline, without interaction with the environment, directly using the limited samples in $\mathcal{D}^{\text{UN}}$ is not effective. Therefore, we leverage the larger set of unlabeled data $\mathcal{D}^{\text{MIX}}$ to enhance the offline training. To achieve this, we first let $\rho^{\text{MIX}}$ be the occupancy measure (or stationary distribution) of the policy represented by unlabeled dataset. We rewrite the expectation over $\rho^{\text{UN}}$ as:

$$\mathbb{E}_{\rho^{\text{UN}}}[r^Q(s,a)] = \sum_{s,a}\rho^{\text{UN}}(s,a)r^Q(s,a) = \sum_{s,a}\rho^{\text{MIX}}(s,a)\tau(s,a)r^Q(s,a) = \mathbb{E}_{\rho^{\text{MIX}}}[\tau(s,a)r^Q(s,a)]$$

Where $\tau(s,a) = \frac{\rho^{\text{UN}}(s,a)}{\rho^{\text{MIX}}(s,a)}$ represents the occupancy ratio between $\rho^{\text{UN}}$ and $\rho^{\text{MIX}}$, we then rewrite the learning objective as follows:

$$\min_Q \left\{ \mathcal{F}(Q) = \mathbb{E}_{\rho^{\text{MIX}}}[\tau(s,a)r^Q(s,a)] - (1-\gamma)\mathbb{E}_{s_0}[V^Q(s_0)] + \psi(r^Q) \right\}$$

In this approach, the expectation $\mathbb{E}_{\rho^{\text{MIX}}}[\tau(s,a)r^Q(s,a)]$ can be empirically approximated using the unlabeled samples from $\mathcal{D}^{\text{MIX}}$, where $\tau(s,a)$ acts as an occupancy correction. This correction allows us to leverage samples from the unlabeled dataset to estimate the expectation over the undesirable policy. A key challenge here is that the occupancy ratio $\tau(s,a)$ is unknown. To address this, we propose estimating the ratio by solving the following implicit maximization problem:

$$\max_{\mu_1,\mu_2:\mathbb{R}^{\mathcal{S}\times\mathcal{A}}\to[0,1]} g(\mu_1,\mu_2) = \mathbb{E}_{\rho^{\text{MIX}}}[\log(\mu_2(s,a) - \mu_1(s,a)\mu_2(s,a))]$$
$$+ \mathbb{E}_{\rho^{\text{UN}}}[\log(\mu_1(s,a) - \mu_1(s,a)\mu_2(s,a))] \tag{7}$$

The above formulation is an extension of the discriminator formulation widely used in prior work (Ho & Ermon, 2016; Kelly et al., 2019). The following proposition theoretically shows that solving 7 will exactly return the occupancy ratio $\tau$.

**Proposition 4.3.** $g(\mu_1,\mu_2)$ *is strictly concave in* $\mu_1,\mu_2$. *Furthermore, let* $\mu_1^*$ *and* $\mu_2^*$ *be the unique optimal solutions to 7, then we have:* $\tau(s,a) = \frac{\mu_1^*(s,a)}{\mu_2^*(s,a)}$.

So, our learning process can be broken down into two steps. In the first step, we learn the occupancy ratios by solving the maximization problems presented in 7. Following this, we optimize the following problem to learn a Q function.

$$\min_Q \left\{ \mathcal{F}(Q) = \mathbb{E}_{\rho^{\text{MIX}}}\left[\frac{\mu_1^*(s,a)}{\mu_2^*(s,a)}r^Q(s,a)\right] - (1-\gamma)\mathbb{E}_{s_0}[V^Q(s_0)] + \psi(r^Q) \right\} \tag{8}$$

It is important to note that we utilize the stationary distribution $\rho^{\text{MIX}}$ (represented by trajectories in the unlabeled dataset) in the objective function in 8. However, thanks to the occupancy correction $\frac{\mu_1^*(s,a)}{\mu_2^*(s,a)}$, the outcome of the training is independent of the quality of the unlabeled policy $\rho^{\text{MIX}}$. This distinguishes our approach from SafeDICE (Jang et al., 2024), where the performance heavily relies on the quality of the unlabeled data.

### 4.3 POLICY EXTRACTION

The Q function obtained from solving the minimization problem in 8 can be used to recover a soft policy $\pi^Q(s,a) = \exp(Q(s,a))/\sum_{a'} \exp(Q(s,a))$, as stated in Proposition 4.2 above. However, this approach may suffer from overestimation, a common issue in offline Q-learning caused by out-of-distribution actions and function approximation errors (Ross et al., 2011). To address this problem and improve policy extraction, we instead propose using weighted behavior cloning (WBC) with the objective: $\max_\pi \left\{ \sum_{s,a} \exp(A(s,a)) \log \pi(a|s) \right\}$, where $A(s,a)$ is the advantage function defined as $A(s,a) = Q(s,a) - V(s)$. It can be shown that solving the WBC problem yields the exact desired soft policy, i.e., $\pi^Q(a|s) = \frac{\exp(Q(s,a))}{\sum_{a'} \exp(Q(s,a))}$, $\forall a, s$, is optimal for the WBC.

## 5 PRACTICAL IMPLEMENTATION

Our algorithm consists of two main steps. The first step involves solving 7 to estimate the occupancy ratio $\tau(s,a) = \frac{\rho^{\text{UN}}(s,a)}{\rho^{\text{MIX}}(s,a)}$. In the second step, we use these ratios to train the Q-function and extract a policy by solving a weighted behavior cloning problem.

In the first step, we construct two networks, $\mu_{\phi_1}(s,a)$ and $\mu_{\phi_2}(s,a) \in [0,1]$, where $\phi_1$ and $\phi_2$ are learnable parameters. We then use samples from $\mathcal{D}^{\text{UN}}$ and $\mathcal{D}^{\text{MIX}}$ to estimate the expectations, leading to the following practical objective:

$$\max_{\phi_1, \phi_2} \quad \widetilde{g}(\phi_1, \phi_2) = \sum_{(s,a) \sim \mathcal{D}^{\text{MIX}}} [\log(\mu_{\phi_2}(s,a) - \mu_{\phi_1}(s,a)\mu_{\phi_2}(s,a))]$$
$$+ \sum_{(s,a) \sim \mathcal{D}^{\text{UN}}} [\log(\mu_{\phi_1}(s,a) - \mu_{\phi_1}(s,a)\mu_{\phi_2}(s,a))] \tag{9}$$

In the second step, after obtaining $\phi_1^*$ and $\phi_2^*$ from the first step, we utilize the following empirical training objective:

$$\min_w \left\{ \widetilde{\mathcal{F}}(w) = \sum_{(s,a,s') \sim \mathcal{D}^{\text{MIX}}} \left[ \frac{\mu_{\phi_1^*}(s,a)}{\mu_{\phi_2^*}(s,a)} r^{Q_w}(s,a,s') + \psi(r^{Q_w}(s,a,s')) \right] \right.$$
$$\left. - (1-\gamma) \sum_{s^0 \sim \mathcal{D}^{\text{MIX}}} V^{Q_w}(s) \right\} \tag{10}$$

where $w$ are the learnable parameters for the Q-network, and the reward function is computed as $r^{Q_w}(s,a,s') = Q_w(s,a) - \gamma V^{Q_w}(s')$, with $V^{Q_w}(s) = \log\left(\sum_{a \sim \mathcal{D}^{\text{MIX}}} \exp(Q_w(s,a))\right)$. Following (Garg et al., 2021), we choose the reward regularizer function $\psi(t) = t - t^2$, i.e., the $\chi^2$-divergence. To extract a policy, we simply solve the weighted BC problem: $\max_\theta \sum_{(s,a) \in \mathcal{D}^{\text{MIX}}} \exp(Q_w(s,a) - V^{Q_w}(s)) \log \pi_\theta(a|s)$, where $\theta$ are learnable parameters of the policy network. Our main algorithm, UNIQ, is summarized as follows, noting that the training of the Q-function $Q_w$ and the updating of the policy network $\pi_\theta$ are performed simultaneously to enhance efficiency.

---

**Algorithm 1 UNIQ**: **UN**desired Demonstrations driven **I**nverse **Q** Learning

---

**Require:** $\mathcal{D}^{UN}, \mathcal{D}^{MIX}, \mu_{\phi_1}, \mu_{\phi_2}, Q_\omega, \pi_\theta, N_\mu, N$
1: # Estimating the occupancy ratios by solving 9
2: **for** certain number of iterations: $i = 1...N_\mu$ **do**
3: $\quad (\phi_1, \phi_2) \leftarrow (\phi_1, \phi_2) + \nabla_{\phi_1, \phi_2} \widetilde{g}(\mu_{\phi_1}, \mu_{\phi_2})$
4: **end for**
5: # train UNIQ
6: **for** certain number of iterations $i = 1...N$ **do**
7: $\quad$ # Update Q function
8: $\quad \omega \leftarrow \omega - \nabla_w \widetilde{\mathcal{F}}(w)$
9: $\quad$ # Update policy via Weighted-BC
10: $\quad \theta \leftarrow \theta + \nabla_\theta \left[ \sum_{(s,a \sim \mathcal{D}^{\text{MIX}})} \exp(Q_w(s,a) - V^{Q_w}(s)) \log \pi_\theta(a|s) \right]$
11: **end for**

---

# 6 EXPERIMENTS

We assess our algorithm in the context of safe RL (or constrained RL) problems, where each state-action pair is associated with a cost value, and the objective is to learn a policy that satisfies certain cost constraints. We define an undesirable trajectory as one where the accumulated cost exceeds a specified threshold. This setting is similar to the one used in SafeDICE (Jang et al., 2024), one of the SOTA algorithms in the context of learning from undesirable demonstrations.

## 6.1 EXPERIMENT SETUP

**Baselines.** We compare our algorithm against several baseline methods. First, we benchmark UNIQ against standard imitation learning algorithms that utilize the entire unlabelled demonstration dataset, specifically BC and IQ-learn, which we refer to as BC-mix and IQ-mix, respectively. Additionally, we benchmark our approach against the SOTA preference-based reinforcement learning algorithm, IPL (Hejna & Sadigh, 2024). Lastly, we include comparisons with DWBC (Xu et al., 2022) and SafeDICE (Jang et al., 2024), both of which are recognized for their ability to learn from undesired demonstrations. More details are provided in the Appendix B.3.

**Environments and Data Generation.** We evaluate our method in four Safety-Gym and two Mujoco environments (Ray et al., 2019; Ji et al., 2023). Following the setup from (Jang et al., 2024), the undesired policy[1] is trained using unconstrained PPO, while the safe policy is trained with SIM (Hoang et al., 2024a). The unlabeled dataset is constructed by combining trajectories from both safe and undesired policies in a specified ratio. The undesired data consists of trajectories that violate the constraint. Detailed information about the policies and datasets is provided in the Appendix B.2.

**Metrics.** The main goal is to train a policy that is safe in the context of constrained RL. Therefore, **an ideal outcome is achieving the lowest possible cost without significantly sacrificing return**. We report the accumulated return and cost for the trained policies, computed based on the last 20 evaluations, with all results summarized across at least 5 training seeds.

**Experimental Concerns.** Throughout the experiments, we aim to address several key questions: (**Q1**) How does UNIQ perform compared to other baseline methods? (**Q2**) How does the presence of undesired demonstrations impact the performance of UNIQ and other baselines? (**Q3**) What happens if the policy is directly extracted from the Q-function, rather than by solving the WBC? (**Q4**) How does the policy learned by UNIQ compare to a policy trained purely from expert demonstrations? (**Q5**) How do UNIQ and other baselines perform when being trained without access to the unlabeled dataset? While (**Q1**) and (**Q2**) are addressed in the main paper, the experiments for the remaining questions are provided in the appendix. The appendix also includes proofs of the theoretical claims made in the main paper, additional details about our experimental setup, and further results such as CVaR cost comparisons, the complete set of experiments for the MuJoCo tasks, comparisons using datasets from the SafeDICE paper, and detailed learning curves.

## 6.2 MAIN COMPARISON ON SAFETY-GYM TASKS

In this section, we aim to evaluate the performance of our method in comparison with the mentioned baselines. We test across three difficulty levels for each environment by varying the amount of safe data (100, 400, and 1600 trajectories) while keeping the number of undesired data fixed at 1600 trajectories. These settings are referred to as *env*-1, 2, 3, respectively. This allows us to examine how the proportion of the desired and undesired demonstrations in the unlabeled dataset impacts the performance of each baseline. The experiment results are shown in Tab. 1.

Overall, as the difficulty increases, both the cost and return of all methods rise. This is primarily because the undesired data typically yields a much higher return than the safe data (except in the Point-Goal task, where the returns of safe and undesired data are close in return). BC-mix and IQ-mix struggle to distinguish between safe and undesired behaviors, leading to poor cost performance. IPL also fails to capture the correct preference, as most of its pairwise comparisons involve undesired-undesired pairs, making it unable to infer the correct preference. DWBC and SafeDICE manage to achieve relatively high returns but fail to match the cost performance of the Safe Policy. Our

---

[1]In the context of safe RL, a desired policy is also referred to as a *safe policy*, while an undesired policy is termed an *unsafe policy*.

|  |  | BC-mix | IQ-Mix | IPL | DWBC | SafeDICE | UNIQ |
|---|---|---|---|---|---|---|---|
| Point-Goal-1 | Return | 26.8±0.1 | 26.7±0.6 | 26.9±0.1 | 26.5±0.2 | 26.6±0.2 | 20.7±0.7 |
|  | Cost | 42.7±3.5 | 43.9±11.9 | 53.7±3.7 | 33.0±3.2 | 36.3±2.9 | **23.5±4.5** |
| Point-Goal-2 | Return | 27.1±0.1 | 27.0±0.4 | 26.9±0.1 | 26.9±0.1 | 27.0±0.1 | 23.4±0.4 |
|  | Cost | 48.8±2.9 | 46.7±10.4 | 52.7±3.4 | 45.8±3.4 | 46.8±3.1 | **27.1±3.0** |
| Point-Goal-3 | Return | 27.1±0.1 | 43.8±29.1 | 26.9±0.1 | 27.1±0.1 | 27.1±0.1 | 26.4±0.2 |
|  | Cost | 51.0±3.4 | 55.6±27.6 | 53.6±3.7 | 50.2±3.6 | 50.7±3.6 | **40.6±3.1** |
| Car-Goal-1 | Return | 32.0±0.6 | 31.7±1.6 | 33.6±0.5 | 28.1±1.2 | 29.8±0.8 | 21.0±0.8 |
|  | Cost | 43.4±4.1 | 42.0±12.5 | 51.1±3.9 | 30.5±3.3 | 36.4±2.9 | **15.4±2.1** |
| Car-Goal-2 | Return | 34.1±0.5 | 34.2±1.5 | 34.7±0.3 | 32.8±0.7 | 33.5±0.7 | 27.9±0.8 |
|  | Cost | 52.0±4.2 | 49.7±13.2 | 54.4±3.7 | 47.4±3.8 | 50.5±4.0 | **31.0±2.8** |
| Car-Goal-3 | Return | 35.2±0.3 | 35.3±0.7 | 35.2±0.2 | 35.0±0.3 | 35.1±0.3 | 34.3±0.4 |
|  | Cost | 56.2±4.8 | 55.2±14.3 | 56.4±4.1 | 55.3±4.0 | 55.5±3.8 | **53.1±4.1** |
| Point-Button-1 | Return | 25.9±1.0 | 26.4±2.2 | 27.0±0.8 | 22.0±0.9 | 23.0±0.9 | 8.8±0.7 |
|  | Cost | 92.9±8.1 | 92.8±23.5 | 114.5±6.4 | 61.5±6.6 | 66.5±6.5 | **12.2±2.7** |
| Point-Button-2 | Return | 29.2±0.7 | 29.8±2.2 | 28.7±0.8 | 28.3±0.8 | 28.8±0.8 | 10.3±0.9 |
|  | Cost | 118.7±7.3 | 123.0±21.4 | 122.1±7.3 | 113.2±7.5 | 114.6±8.7 | **19.1±3.0** |
| Point-Button-3 | Return | 30.6±0.7 | 30.8±1.9 | 29.6±0.6 | 30.4±0.6 | 30.3±0.7 | 14.9±1.1 |
|  | Cost | 130.9±9.0 | 129.2±23.7 | 129.9±7.3 | 131.5±8.8 | 128.6±8.7 | **55.5±7.6** |
| Car-Button-1 | Return | 14.1±1.2 | 14.3±3.6 | 17.6±1.5 | 10.1±1.1 | 11.8±1.3 | 2.3±0.4 |
|  | Cost | 132.3±12.7 | 126.6±38.3 | 165.7±14.4 | 101.0±14.3 | 116.2±13.1 | **35.9±5.4** |
| Car-Button-2 | Return | 21.0±1.3 | 20.6±3.6 | 22.9±1.1 | 18.7±1.4 | 21.4±1.5 | 5.1±0.7 |
|  | Cost | 191.4±13.9 | 189.4±45.8 | 209.8±12.2 | 178.0±14.3 | 198.1±15.4 | **65.8±10.1** |
| Car-Button-3 | Return | 24.8±0.9 | 24.3±3.4 | 25.2±0.9 | 24.1±1.1 | 24.7±1.0 | 14.0±1.5 |
|  | Cost | 223.7±10.8 | 229.5±41.6 | 230.4±11.2 | 220.9±12.8 | 232.5±10.9 | **144.0±15.2** |

Table 1: Comparison results for Safety-Gym tasks.

method consistently achieves the lowest cost across all experiments. However, in the Point-Button and Car-Button tasks, the return for our method is lower, as it avoids undesired actions, leaving no high-return options to pursue. The detailed learning curves are shown in Appendix D.1.

## 6.3 MUJOCO VELOCITY BENCHMARKS

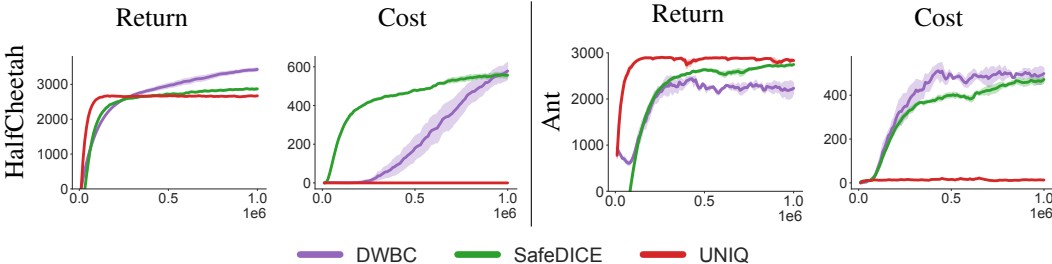

Figure 2: Learning curves of UNIQ, DWBC and SafeDICE in Mujoco-velocity tasks.

We test our method with two MuJoCo velocity tasks: Cheetah and Ant. We keep the same size of the unlabelled dataset as difficulty level 2 of the Safety-Gym experiments (Section 6.2), which is composed of 400 trajectories from the desired data and 1600 trajectories from the undesired data. Moreover, due to the nature of the environment, we only require a smaller number of labeled undesired datasets, which is 5. Detailed results are shown in Figure 2, and a more detailed comparison of the MuJoCo domain is provided in Appendix C.8. In general, UNIQ outperforms other baselines in both tasks with a significantly lower cost.

## 6.4 ABLATION STUDY - PERFORMANCE WITH DIFFERENT SIZES OF UNDESIRED DATASET

In this experiment, we evaluate our method using varying sizes of the undesired dataset (25, 50, 100, 200, 300, and 500 trajectories), while keeping the unlabelled dataset fixed (400 desired and 1600 undesired trajectories) across two Safety-Gym environments. The results, reported in Figure 3, include the return, cost, and CVaR 10% cost for each undesired dataset size, where CVaR 10% cost is the mean cost of the worst 10% runs in the evaluation. The training curves are shown in the Appendix D.2. Here, BC-safe refers to Behavioral Cloning with only the desired (or expert) demonstrations, which serves as the highest safety performance benchmark. In general, increasing the size of the undesired dataset tends to reduce the cost for all approaches, and UNIQ shows the greatest effect in utilizing the undesired data. Interestingly, UNIQ is able to achieve a lower cost than BC-safe, which can be explained by the fact that the main goal of UNIQ is to avoid the undesired (i.e., high-cost) demonstrations, while BC-safe lacks this avoidance capability.

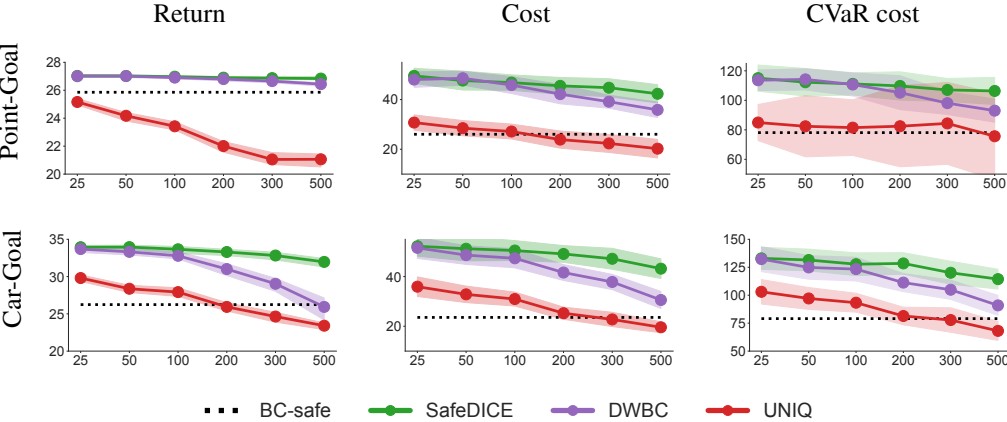

Figure 3: Comparison results for different sizes of the undesirable dataset.

## 7 CONCLUSION, FUTURE WORK, AND BROADER IMPACTS

**Conclusion.** We have developed UNIQ, a principled framework based on inverse Q-learning to facilitate learning from undesirable demonstrations. Our algorithm can be seen as a reverse version of standard imitation learning frameworks, where the goal is to maximize the statistical distance between the learning policy and the undesirable policy. UNIQ requires minimal hyper-parameter tuning, as it does not introduce any additional hyperparameters beyond those typically used in inverse Q-learning algorithms. Moreover, it demonstrates superior performance in producing safe policies in several safe reinforcement learning experiments, outperforming other baseline methods.

**Limitations and Future work.** There are several aspects that have not been addressed in this paper, as they are too significant to be fully explored here. For instance, we generally assume the presence of only one set of undesirable demonstrations, whereas in practice, multiple datasets of varying quality could be leveraged to enhance the training. Additionally, each undesirable trajectory may not be undesirable in its entirety, as it could contain some good actions. Extracting the good parts from undesirable demonstrations could improve sample efficiency but introduces new challenges that warrant further investigation. Another open question is how to extend the framework to multi-agent settings, which would be both relevant and interesting to explore in future research.

**Broader Impact.** Beyond the standard impacts of imitation learning, our work is particularly useful in scenarios where undesirable demonstrations are richer, clearer, or more reliable than desirable ones. This is especially valuable in critical applications like healthcare and autonomous driving, where avoiding harmful actions is essential. However, our approach also carries potential negative impacts. If not carefully designed, the system might unintentionally reinforce undesirable behaviors or learn harmful actions from poorly curated data. Misinterpreting bad demonstrations could result in unintended consequences, particularly in safety-critical contexts. Additionally, there is a risk that malicious actors could misuse this framework to deliberately train AI systems with harmful or unethical behaviors.

## ETHICAL STATEMENT

Our work on the UNIQ framework addresses learning from undesirable demonstrations. This research holds potential for impactful real-world applications, including areas such as autonomous systems, healthcare, and robotics, where avoiding harmful behaviors is crucial. However, it is important to acknowledge the ethical implications and risks associated with our work.

While the UNIQ framework aims to learn safe policies by avoiding undesirable behaviors, there is a potential risk that the algorithm could be misused in ways that reinforce unintended or harmful actions if trained on poorly curated data. For example, in scenarios where undesirable demonstrations are not clearly defined, the model could inadvertently reinforce biased or harmful behaviors. Furthermore, there is the risk of deploying the framework in environments where safety-critical decisions are made without sufficient validation or human oversight, leading to unintended consequences.

To mitigate these risks, we emphasize the importance of using well-curated datasets that accurately represent undesirable behaviors and conducting thorough testing in controlled environments before applying the system to real-world scenarios. Moreover, we encourage transparency and collaboration with domain experts to ensure the framework is used responsibly, particularly in safety-critical applications such as healthcare and autonomous driving. Finally, we advocate for the inclusion of human oversight in the deployment of policies learned using the UNIQ framework to ensure ethical and safe outcomes.

## REPRODUCIBILITY STATEMENT

To ensure the reproducibility of our work, we have submitted the source code for the UNIQ framework along with the datasets used to generate the experimental results reported in this paper (this source code and datasets will be made publicly available if the paper gets accepted). We also provide comprehensive details of our algorithm in the appendix, including implementation details and key steps required to reproduce the experimental outcomes. Additionally, we have included the hyper-parameter configurations used in all experiments, ensuring that others can replicate the results under the same conditions. We encourage the research community to build on our work and test the UNIQ framework across different environments to validate and extend the findings presented in this paper.

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

APPENDIX

The appendix contains the following details:

**Missing Proofs:** Refer to Appendix A for proofs omitted from the main paper.

**Experimental Details:** We provide information on:

- Baseline implementation (Appendix B.3)
- Hyper-parameter selection for each task domain (Appendix B.4)
- Task descriptions (Appendix B.1)
- Generation of undesirable and unlabelled datasets (Appendix B.2)

**Additional Experiments:** We address the following remaining questions:

- (**Q3**) What happens if the policy is directly extracted from the Q-function, rather than by solving the WBC? (Appendix C.2)
- (**Q4**) How does the policy learned by UNIQ compare to a policy trained purely from expert demonstrations? (Appendix C.3)
- (**Q5**) How do UNIQ and other baselines perform when being trained without access to the unlabeled dataset? (Appendix C.4)

Additionally, we present experiments demonstrating the performance of UNIQ and other baseline methods on the dataset from the safeDICE paper (see Appendix C.9) and provide comparison results using CVaR costs (see Appendix C.1).

**Supplementary Learning Curves:** We provide learning curves to for the results reported in:

- Table 1 (Appendix D.1)
- Figure 3 (Appendix D.2)

CONTENTS

## A  MISSING PROOFS

### A.1  PROOF OF PROPOSITION 4.1

**Proposition.**  *For a non-restricted feasible set of the reward function:*

$$\min_r \min_\pi \{L(\pi, r)\} = -\max_\pi \{d_\psi(\rho^\pi, \rho^{\mathrm{UN}}) - H(\pi)\}$$

*where $d_\psi(\rho^\pi, \rho^{\mathrm{UN}}) = \psi^*(\rho^\pi - \rho^{\mathrm{UN}})$, and $\psi^*$ is the convex conjugate of the convex function $\psi$, i.e., $\psi^*(t) = \sup_z(tz - \psi(z))$.*

*Proof.*  We write the objective function as

$$L(\pi, r) = \sum_{s,a} r(s, a) \left(\rho^{\mathrm{UN}}(s, a) - \rho^\pi(s, a)\right) + \psi(r) - H(\pi)$$

So we can write the minimization of $L(\pi, r)$ over $r$ as:

$$\min_r L(\pi, r) = H(\pi) + \min_r \ \sum_{s,a} r(s, a) \left(\rho^{\mathrm{UN}}(s, a) - \rho^\pi(s, a)\right) + \psi(r)$$

$$= H(\pi) - \max_r \left\{ \sum_{s,a} \left\{ r(s, a) \left(\rho^\pi(s, a) - \rho^{\mathrm{UN}}(s, a)\right) \right\} - \psi(r) \right\}$$

Since the feasible set of the reward function $r$ is unrestricted, we know that

$$\max_r \left\{ \langle r, \rho^\pi - \rho^{\mathrm{UN}} \rangle - \psi(r) \right\} = \psi^*(\rho^\pi - \rho^{\mathrm{UN}})$$

which allows us to rewrite the training problem as:

$$\min_r \min_\pi \{L(\pi, r)\} = \min_\pi \left\{ H(\pi) - \psi^*\left(\rho^\pi - \rho^{\mathrm{UN}}\right) \right\}$$

$$= -\max_\pi \left\{ \psi^*(\rho^\pi - \rho^{\mathrm{UN}}) - H(\pi) \right\}$$

$$= -\max_\pi \{d_\psi(\rho^\pi, \rho^{\mathrm{UN}}) - H(\pi)\}$$

We obtain the desired equation and complete the proof.  $\square$

### A.2  PROOF OF PROPOSITION 4.2

**Proposition.**  *The following statements hold:*

*(i) The function $L(\pi, Q)$ is convex in $\pi$ and the problem $\min_\pi L(\pi, Q)$ has a unique optimal solution at $\pi^Q(s, a) = \frac{\exp(Q(s,a))}{\sum_{a'} \exp(Q(s,a'))}$.*

*(ii) The learning objective function can be simplified as:*

$$\min_Q \min_\pi \{L(\pi, Q)\} = \min_Q \left\{ \mathcal{F}(Q) = \mathbb{E}_{\rho^{\mathrm{UN}}}[r^Q(s, a)] - (1 - \gamma)\mathbb{E}_{s_0}[V^Q(s_0)] + \psi(r^Q) \right\}$$

*where $V^Q(s) = \log\left(\sum_a \exp(Q(s, a))\right)$ and $r^Q(s, a) = Q(s, a) - \gamma\mathbb{E}_{s' \sim P(\cdot|s,a)} V^Q(s')$.*

*Proof.*  We first express the second and third terms of the objective function $L(\pi, Q)$ in 5 as:

$$\mathbb{E}_{\rho_\pi}[\mathcal{T}^\pi[Q](s, a)] + H(\pi) = \mathbb{E}_{\rho_\pi}[Q(s, a) - \gamma\mathbb{E}_{s'}[V^\pi(s')]] - \mathbb{E}_{\rho^\pi}[\log \pi(s, a)]$$

$$= \mathbb{E}_{\rho_\pi}[Q(s, a) - \log \pi(s, a) - \gamma\mathbb{E}_{s'}[V^\pi(s')]] = \mathbb{E}_{\rho_\pi}[V(s) - \gamma\mathbb{E}_{s' \sim P(\cdot|s,a)}[V^\pi(s')]]$$

$$= (1 - \gamma)\mathbb{E}_{s_0 \sim P_0}[V^\pi(s_0)].$$

Thus, the objective function becomes:

$$L(\pi, Q) = \mathbb{E}_{\rho^{\mathrm{UN}}}[Q(s, a) - \gamma\mathbb{E}_{s'}[V^\pi(s')]] - (1 - \gamma)\mathbb{E}_{s_0 \sim P_0}[V^\pi(s_0)] + \sum_{s,a} \psi(Q(s, a) - \gamma\mathbb{E}_{s'}[V^\pi(s')]).$$

We now observe that $V^\pi(s) = \mathbb{E}_{a \sim \pi(a|s)}[Q(s,a) - \log \pi(a|s)]$ is concave in $\pi$. Therefore, both terms $\mathbb{E}_{\rho^{\mathrm{UN}}}[Q(s,a) - \gamma \mathbb{E}_{s'}[V^\pi(s')]]$ and $-(1-\gamma)\mathbb{E}_{s_0 \sim P_0}[V^\pi(s_0)]$ are convex in $\pi$. Additionally, since $\psi(t)$ is convex and non-increasing in $t$, and $Q(s,a) - \gamma \mathbb{E}_{s'}[V^\pi(s')]$ is convex in $\pi$, each function $\psi(Q(s,a) - \gamma \mathbb{E}_{s'}[V^\pi(s')])$ is convex in $\pi$. Thus, combining all terms, we conclude that $L(\pi, Q)$ is convex in $\pi$.

Furthermore, each term $Q(s,a) - \gamma \mathbb{E}_{s'}[V^\pi(s')]$, $-(1-\gamma)\mathbb{E}_{s_0 \sim P_0}[V^\pi(s_0)]$, and $\psi(Q(s,a) - \gamma \mathbb{E}_{s'})$ strictly decreases in $V^\pi$, implying that the minimization of $L(\pi, Q)$ over $\pi$ is achieved when $V^\pi(s)$ is maximized for all $s$. Since $V^\pi(s)$ is strictly concave in $\pi$, maximizing $V^\pi(s)$ over $\pi$ has a unique optimal solution:

$$\pi^Q(a|s) = \frac{\exp(Q(s,a))}{\sum_a \exp(Q(s,a))}.$$

This validates part (i) of the theorem.

For part (ii), we observe that the problem $\max_\pi V^\pi(s)$ has the optimal solution $\pi^Q$ as shown above, and the optimal value is:

$$\max_\pi V^\pi(s) = \max_\pi \left\{ \sum_a \pi(a|s)Q(s,a) - \pi(a|s)\log\pi(a|s) \right\}$$

$$= \log\left(\sum_a \exp(Q(s,a))\right) \stackrel{\text{def}}{=} V^Q(s).$$

This directly leads to:

$$\min_Q \min_\pi \{L(\pi, Q)\} = \min_Q \left\{ \mathcal{F}(Q) = \mathbb{E}_{\rho^{\mathrm{UN}}}[r^Q(s,a)] - (1-\gamma)\mathbb{E}_{s_0}[V^Q(s_0)] + \psi(r^Q) \right\},$$

as required.

$\square$

## A.3 Proof of Proposition 4.3

**Proposition:** $g(\mu_1, \mu_2)$ is strictly concave in $\mu_1, \mu_2$. Furthermore, let $\mu_1^*$ and $\mu_2^*$ be the unique optimal solutions to 7, then we have:

$$\tau(s,a) = \frac{\mu_1^*(s,a)}{\mu_2^*(s,a)}.$$

*Proof.* We begin by expressing the objective function as follows:

$$g(\mu_1, \mu_2) = \mathbb{E}_{\rho^{\mathrm{MIX}}}[\log(1 - \mu_1(s,a))] + \mathbb{E}_{\rho^{\mathrm{UN}}}[\log(\mu_1(s,a))] +$$
$$\mathbb{E}_{\rho^{\mathrm{UN}}}[\log(1 - \mu_2(s,a))] + \mathbb{E}_{\rho^{\mathrm{MIX}}}[\log(\mu_2(s,a))].$$

Each term, $\log(1 - \mu_1(s,a))$, $\log(\mu_1(s,a))$, $\log(1 - \mu_2(s,a))$, and $\log(\mu_2(s,a))$, is strictly concave in $\mu_1$ and $\mu_2$. Therefore, the objective function $g(\mu_1, \mu_2)$ is also strictly concave in $\mu_1$ and $\mu_2$. To find the unique optimal solution to this problem, we compute the first-order derivatives of $g(\mu_1, \mu_2)$ with respect to $\mu_1$ and $\mu_2$ and set them to zero:

$$\frac{\rho^{\mathrm{UN}}(s,a)}{\mu_1(s,a)} - \frac{\rho^{\mathrm{MIX}}(s,a)}{1 - \mu_1(s,a)} = 0,$$

$$\frac{\rho^{\mathrm{MIX}}(s,a)}{\mu_2(s,a)} - \frac{\rho^{\mathrm{UN}}(s,a)}{1 - \mu_2(s,a)} = 0.$$

This leads to the following system of equations:

$$\rho^{\mathrm{UN}}(s,a)(1 - \mu_1(s,a)) = \rho^{\mathrm{MIX}}(s,a)\mu_1(s,a),$$

$$\rho^{\mathrm{MIX}}(s,a)(1 - \mu_2(s,a)) = \rho^{\mathrm{UN}}(s,a)\mu_2(s,a).$$

Solving this system yields the unique solutions:

$$\mu_1^*(s, a) = \frac{\rho^{\text{UN}}(s, a)}{\rho^{\text{UN}}(s, a) + \rho^{\text{MIX}}(s, a)},$$

$$\mu_2^*(s, a) = \frac{\rho^{\text{MIX}}(s, a)}{\rho^{\text{UN}}(s, a) + \rho^{\text{MIX}}(s, a)}.$$

We observe that both $\mu_1^*(s, a)$ and $\mu_2^*(s, a)$ lie in the interval $(0, 1)$, confirming that they are the unique solutions that maximize $g(\mu_1, \mu_2)$. Moreover, we validate the equality:

$$\frac{\mu_1^*(s, a)}{\mu_2^*(s, a)} = \frac{\rho^{\text{UN}}(s, a)}{\rho^{\text{MIX}}(s, a)}.$$

$\square$

## B EXPERIMENT SETTINGS

### B.1 ENVIRONMENT DETAILS

#### B.1.1 SAFE-GYM

Safe-Gym is a collection of reinforcement learning environments designed with a focus on safety, built on top of the OpenAI Gym framework. It introduces constraints that simulate safety-critical scenarios commonly encountered in real-world applications. In Safe-Gym, agents are rewarded for completing task-specific objectives but face penalties for violating safety constraints, such as surpassing speed limits, colliding with obstacles, or entering restricted zones. These constraints enable Safe-Gym to replicate environments where safety is paramount, including robotic navigation in congested areas, autonomous vehicle control, and industrial automation. The environment features two types of agents: Point (an easy agent) and Car (a more challenging agent), as well as two types of tasks: Goal (easy) and Button (hard). Additionally, the environment dynamics change with each new episode, introducing variability and increasing the complexity of the tasks. Illustrations of these tasks are shown in Figure 4.

| Point-Goal | Car-Goal | Point-Button | Car-Button |

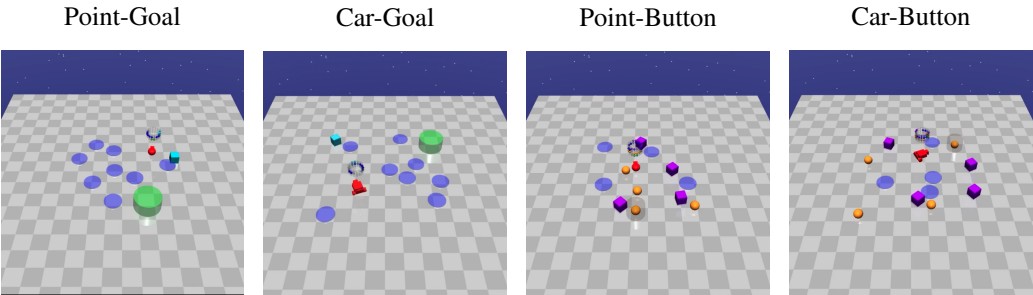

Figure 4: Safety-gym environments.

#### B.1.2 MUJOCO-VELOCITY

Mujoco-Velocity is a specialized environment within the Mujoco physics simulation suite, focusing on controlling the velocity of two specific agents: Cheetah and Ant. These agents must complete locomotion tasks while adhering to safety constraints on their speed. The goal is to balance task performance with maintaining safe velocity limits. For instance, Cheetah must run as fast as possible while staying within predefined speed bounds to avoid penalties, mimicking real-world scenarios where exceeding speed limits can cause system failure or unsafe operations. Similarly, Ant must navigate through its environment without violating velocity constraints, ensuring stability and safety. The illustrations are shown in Figure 5.

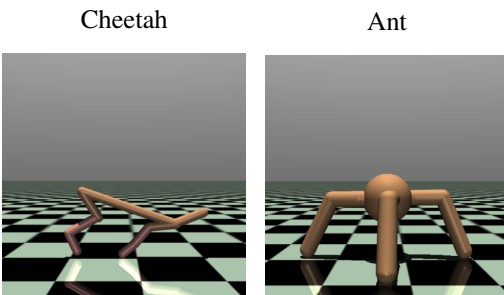

Cheetah      Ant

Figure 5: Mujoco-velocity environments.

## B.2 DATASET GENERATION DETAILS

In this paper, we tackle the problem of **Offline *Reverse* Imitation Learning** by utilizing two datasets:

- Unlabelled dataset $\mathcal{D}^{MIX}$: A large dataset comprising both desired and undesired demonstrations, reflecting real-world data (e.g., chat conversations, driving behaviors, treatment decisions, etc.).

- Undesired dataset $\mathcal{D}^{UN}$: A smaller, accurated dataset containing demonstrations that exhibit behaviors we aim to avoid.

We simulate this scenario using the Safety-gym and Mujoco-velocity environments. First, we train both unconstrained and constrained policies using PPO and SIM (Hoang et al., 2024a) (SIM is an incremental method that achieves significantly higher returns while incurring lower costs compared to other safe methods). We then collect the training datasets as follows:

- Unlabelled dataset $\mathcal{D}^{MIX}$: We roll out the constrained and unconstrained policies, mixing them at ratios of $(1:1, 1:4, 1:16)$ to progressively increase task difficulty.

- Undesired dataset $\mathcal{D}^{UN}$: We roll out the unconstrained policy, gathering trajectories that violate the constraint.

### B.2.1 SAFETY-GYM DATASET DETAILED QUALITY

The details information of Safety-Gym datasets are shown in Table 2.

|  | Point-Goal | Car-Goal | Point-Button | Car-Button |
|---|---|---|---|---|
| Mean Unconstrained return | $26.8 \pm 1.1$ | $35.2 \pm 2.1$ | $30.0 \pm 5.3$ | $25.5 \pm 8.4$ |
| Mean Constrained return | $25.3 \pm 2.1$ | $25.7 \pm 6.6$ | $15.6 \pm 7.3$ | $4.1 \pm 5.2$ |
| Mean undesired return | $26.8 \pm 1.1$ | $34.7 \pm 1.8$ | $29.5 \pm 5.3$ | $25.3 \pm 8.0$ |
| Mean Unconstrained cost | $57.8 \pm 38.9$ | $61.9 \pm 48.1$ | $129.9 \pm 77.3$ | $245.5 \pm 115.9$ |
| Mean Constrained cost | $22.3 \pm 28.0$ | $21.0 \pm 31.0$ | $24.4 \pm 34.1$ | $51.6 \pm 78.2$ |
| Mean undesired cost | $80.9 \pm 23.5$ | $90.9 \pm 32.2$ | $139.0 \pm 80.4$ | $251.2 \pm 105.7$ |
| Cost Threshold | $25.0$ | $25.0$ | $25.0$ | $50.0$ |

Table 2: Safety-Gym expert policies performance.

### B.2.2 MUJOCO-VELOCITY DATASET DETAILED QUALITY

The details information of safety-gym datasets are shown in Table 3.

|                             | Cheetah           | Ant                |
|-----------------------------|-------------------|--------------------|
| Mean Unconstrained return   | $3027.4 \pm 400.6$ | $2972.2 \pm 1020.0$ |
| Mean Constrained return     | $2751.2 \pm 11.6$ | $2830.0 \pm 145.5$ |
| Mean undesired return       | $3010.3 \pm 424.2$ | $2996.5 \pm 991.5$ |
| Mean Unconstrained cost     | $626.7 \pm 95.0$  | $624.0 \pm 231.0$  |
| Mean Constrained cost       | $14.1 \pm 4.5$    | $18.7 \pm 4.4$     |
| Mean undesired cost         | $622.5 \pm 99.7$  | $629.6 \pm 221.4$  |
| Cost Threshold              | 25.0              | 25.0               |

Table 3: Mujoco-velocity expert policies performance.

### B.3 BASELINE IMPLEMENTATION DETAILS

#### B.3.1 BC

We use the orginal BC objective:

$$\min_{\pi} -\mathbb{E}_{s,a\sim\mathcal{D}} \log \pi(a|s) \tag{11}$$

#### B.3.2 IQLEARN

We use the official implementation of IQ-Learn from the DualRL paper (Sikchi et al., 2024), available through this link. Moreover, to improve stability in offline training, we modify the actor update part to be the same as in UNIQ (Algorithm 1). The performance comparison between the two versions is provided in Appendix C.2.

#### B.3.3 IPL

We use the official implementation of IPL (Hejna & Sadigh, 2024) from this link. The only difference between our setting and IPL is the pairwise dataset. We create pairwise comparisons from the unlabelled dataset and the undesired dataset and train the Q-function with the following new loss function:

$$P_{Q^\pi}[\sigma^{MIX} > \sigma^{UN}] = \frac{\exp \sum_t (\mathcal{T}^\pi Q)(s_t^{MIX}, a_t^{MIX})}{\exp \sum_t (\mathcal{T}^\pi Q)(s_t^{MIX}, a_t^{MIX}) + \exp \sum_t (\mathcal{T}^\pi Q)(s_t^{UN}, a_t^{UN})},$$

where:

$$(\mathcal{T}^\pi Q)(s,a) = Q(s,a) - \gamma \mathbb{E}_{s'}[V^\pi(s')].$$

#### B.3.4 DWBC

We use the official implementation of DWBC (Xu et al., 2022) from this link. We modify the algorithm to train a discriminator that assigns 1 to the undesired dataset $\mathcal{D}^{UN}$ and 0 to the unlabelled dataset $\mathcal{D}^{MIX}$ while keeping the Positive Unlabeled learning technique from the paper:

$$\min_{\theta} \eta \mathbb{E}_{(s,a)\sim\mathcal{D}^{UN}} \left[ -\log d_\theta(s,a,\log \pi) \right]$$
$$+ \mathbb{E}_{(s,a)\sim\mathcal{D}^{MIX}} \left[ -\log \left( 1 - d_\theta(s,a,\log \pi) \right) \right]$$
$$- \eta \mathbb{E}_{(s,a)\sim\mathcal{D}^{UN}} \left[ -\log \left( 1 - d_\theta(s,a,\log \pi) \right) \right].$$

We then learn the policy by optimizing:

$$\min_{\psi} -\mathbb{E}_{(s,a)\sim\mathcal{D}^{MIX}} \left[ \left( \frac{1}{d_\theta(s,a)} - 1 \right) \log \pi_\psi(a|s) \right].$$

#### B.3.5 SAFEDICE

We use the official implementation of SafeDICE (Jang et al., 2024) from this link. As the algorithm has been designed to solve this problem, we do not make any further modifications.

### B.4 HYPER-PARAMETER SELECTION

For fair comparison, we keep the same basic hyper-parameters across all the baselines which are detailed as follow for Safety-gym and Mujoco-velocity tasks:

| HYPER PARAMETER | SAFETY-GYM | MUJOCO-VELOCITY |
|---|---|---|
| ACTOR NETWORK | [256,256,256] | [256,256] |
| CRITIC NETWORK | [256,256,256] | [256,256] |
| TRAINING STEP | 1,000,000 | 1,000,000 |
| GAMMA | 0.99 | 0.99 |
| LR ACTOR | 0.0001 | 0.0001 |
| LR CRITIC | 0.0003 | 0.0003 |
| LR DISCRIMINATOR | 0.0001 | 0.0001 |
| BATCH SIZE | 256 | 256 |
| SOFT UPDATE CRITIC FACTOR | 0.005 | 0.005 |

Table 4: Hyper parameters.

Lastly, we apply state normalization for Mujoco-velocity datasets as follow:

$$s_{\text{normalized}} = \frac{s - \mu}{\sigma}$$

Where:

$$\mu = \frac{1}{|\mathcal{D}|} \sum_{s' \in \mathcal{D}} s'$$

$$\sigma = \sqrt{\frac{1}{|\mathcal{D}|} \sum_{s' \in \mathcal{D}} (s' - \mu)^2}$$

# C ADDITIONAL EXPERIMENTS

## C.1 CVAR 10% COST OF SAFETY-GYM COMPARISON

We also report the CVaR 10% cost, supporting the result of the Table 1 with CVaR is the mean of 10% highest in cost trajectories during the evaluation process. The full results are shown in Table 5.

| | | BC-mix | IQ-Mix | IPL | DWBC | SafeDICE | UNIQ |
|---|---|---|---|---|---|---|---|
| Point-Goal-1 | Return | 26.8±0.1 | 26.7±0.6 | 26.9±0.1 | 26.5±0.2 | 26.6±0.2 | 20.7±0.7 |
| | Cost | 42.7±3.5 | 43.9±11.9 | 53.7±3.7 | 33.0±3.2 | 36.3±2.9 | **23.5±4.5** |
| | CVaR | 106.3±10.3 | 103.8±54.4 | 119.3±7.8 | **90.1±9.6** | 94.8±10.9 | 101.0±38.0 |
| Point-Goal-2 | Return | 27.1±0.1 | 27.0±0.4 | 26.9±0.1 | 26.9±0.1 | 27.0±0.1 | 23.4±0.4 |
| | Cost | 48.8±2.9 | 46.7±10.4 | 52.7±3.4 | 45.8±3.4 | 46.8±3.1 | **27.1±3.0** |
| | CVaR | 115.4±7.7 | 105.3±24.1 | 117.9±8.2 | 110.4±7.8 | 111.0±7.6 | **85.9±18.9** |
| Point-Goal-3 | Return | 27.1±0.1 | 43.8±29.1 | 26.9±0.1 | 27.1±0.1 | 27.1±0.1 | 26.4±0.2 |
| | Cost | 51.0±3.4 | 55.6±27.6 | 53.6±3.7 | 50.2±3.6 | 50.7±3.6 | **40.6±3.1** |
| | CVaR | 116.8±7.6 | 106.6±21.7 | 119.5±8.3 | 115.7±7.8 | 116.6±8.1 | **102.6±10.3** |
| Car-Goal-1 | Return | 32.0±0.6 | 31.7±1.6 | 33.6±0.5 | 28.1±1.2 | 29.8±0.8 | 21.0±0.8 |
| | Cost | 43.4±4.1 | 42.0±12.5 | 51.1±3.9 | 30.5±3.3 | 36.4±2.9 | **15.4±2.1** |
| | CVaR | 117.5±10.9 | 103.4±30.5 | 129.4±9.7 | 90.5±10.3 | 103.8±9.9 | **62.1±8.7** |
| Car-Goal-2 | Return | 34.1±0.5 | 34.2±1.5 | 34.7±0.3 | 32.8±0.7 | 33.5±0.7 | 27.9±0.8 |
| | Cost | 52.0±4.2 | 49.7±13.2 | 54.4±3.7 | 47.4±3.8 | 50.5±4.0 | **31.0±2.8** |
| | CVaR | 132.8±10.6 | 114.5±29.0 | 134.9±8.7 | 123.2±10.6 | 128.5±10.2 | **93.3±8.4** |
| Car-Goal-3 | Return | 35.2±0.3 | 35.3±0.7 | 35.2±0.2 | 35.0±0.3 | 35.1±0.3 | 34.3±0.4 |
| | Cost | 56.2±4.8 | 55.2±14.3 | 56.4±4.1 | 55.3±4.0 | 55.5±3.8 | **53.1±4.1** |
| | CVaR | 140.3±12.3 | **127.9±35.0** | 139.5±10.9 | 139.5±10.9 | 138.8±9.9 | 134.8±9.4 |
| Point-Button-1 | Return | 25.9±1.0 | 26.4±2.2 | 27.0±0.8 | 22.0±0.9 | 23.0±0.9 | 8.8±0.7 |
| | Cost | 92.9±8.1 | 92.8±23.5 | 114.5±6.4 | 61.5±6.6 | 66.5±6.5 | **12.2±2.7** |
| | CVaR | 241.7±30.7 | 209.2±64.6 | 277.2±27.3 | 182.8±24.8 | 187.3±22.4 | **61.5±15.6** |
| Point-Button-2 | Return | 29.2±0.7 | 29.8±2.2 | 28.7±0.8 | 28.3±0.8 | 28.8±0.8 | 10.3±0.9 |
| | Cost | 118.7±7.3 | 123.0±21.4 | 122.1±7.3 | 113.2±7.5 | 114.6±8.7 | **19.1±3.0** |
| | CVaR | 273.6±30.4 | 263.6±97.7 | 281.6±29.0 | 267.4±27.0 | 270.1±33.8 | **81.1±14.9** |
| Point-Button-3 | Return | 30.6±0.7 | 30.8±1.9 | 29.6±0.6 | 30.4±0.6 | 30.3±0.7 | 14.9±1.1 |
| | Cost | 130.9±9.0 | 129.2±23.7 | 129.9±7.3 | 131.5±8.8 | 128.6±8.7 | **55.5±7.6** |
| | CVaR | 273.6±30.4 | 267.9±74.5 | 293.5±24.6 | 267.4±27.0 | 270.1±33.8 | **81.1±14.9** |
| Car-Button-1 | Return | 14.1±1.2 | 14.3±3.6 | 17.6±1.5 | 10.1±1.1 | 11.8±1.3 | 2.3±0.4 |
| | Cost | 132.3±12.7 | 126.6±38.3 | 165.7±14.4 | 101.0±14.3 | 116.2±13.1 | **35.9±5.4** |
| | CVaR | 387.0±44.2 | 334.6±125.3 | 430.4±47.2 | 345.5±49.7 | 363.8±47.4 | **183.6±30.7** |
| Car-Button-2 | Return | 21.0±1.3 | 20.6±3.6 | 22.9±1.1 | 18.7±1.4 | 21.4±1.5 | 5.1±0.7 |
| | Cost | 191.4±13.9 | 189.4±45.8 | 209.8±12.2 | 178.0±14.3 | 198.1±15.4 | **65.8±10.1** |
| | CVaR | 449.2±45.8 | 421.1±136.6 | 482.7±48.7 | 451.3±43.7 | 465.3±36.5 | **294.6±46.7** |
| Car-Button-3 | Return | 24.8±0.9 | 24.3±3.4 | 25.2±0.9 | 24.1±1.1 | 24.7±1.0 | 14.0±1.5 |
| | Cost | 223.7±10.8 | 229.5±41.6 | 230.4±11.2 | 220.9±12.8 | 232.5±10.9 | **144.0±15.2** |
| | CVaR | 474.5±39.3 | 461.9±136.4 | 500.3±53.3 | 473.1±36.6 | 492.4±48.8 | **432.0±44.1** |

Table 5: Full comparison results in Return, Cost, and CVaR 10%.

## C.2 COMPARISON WITH POLICIES DIRECTLY EXTRACTED FROM Q-FUNCTIONS

As our method use a different policy update compared to the original policy update of IQlearn (Garg et al., 2021), we want to show the reason why we modify policy update method into **Weighted-BC**. Here, we keep the same data amount of level 2 for all four safety-gym environments. the comparison results are shown in Figure 6. From 5 training seeds, the result showing that the performance of **direct extraction** (original policy update of IQlearn) are not stable across different task while our new **Weighted BC** version have better stability.

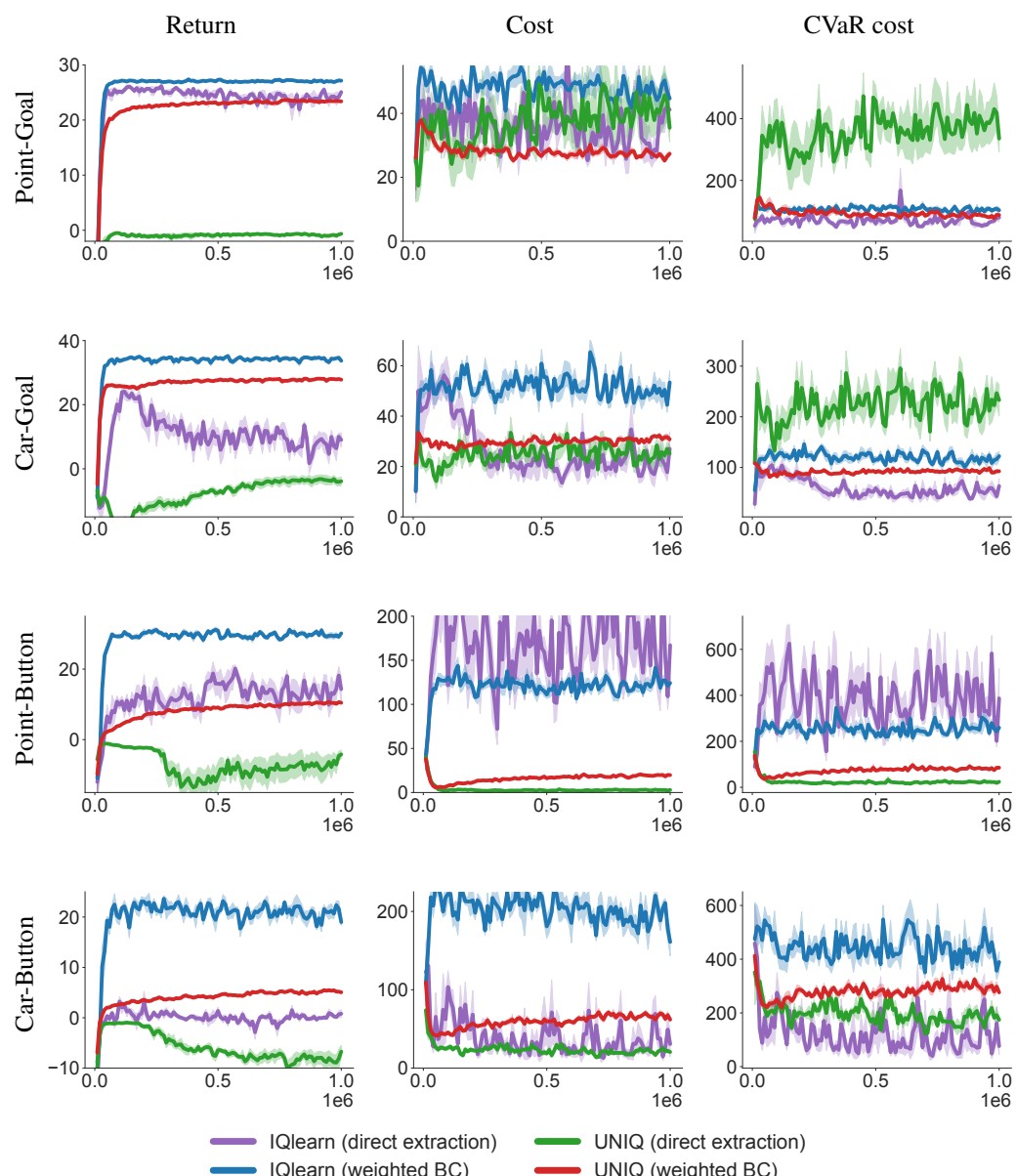

Figure 6: We compared the performance of Q-inference (the original policy update of IQlearn) with our Weighted-BC update. Despite averaging the results over 5 training seeds, the learning curves for direct extraction runs showed significantly more fluctuation compared to those of our Weighted-BC runs.

## C.3 COMPARISON WITH POLICIES LEARNED FROM EXPERT (OR DESIRED) DEMONSTRATIONS

We aim to evaluate the performance when learning directly from **desired demonstrations** (or expert demonstrations), without the noise of unlabelled undesirable trajectories in the unlabelled dataset. This can be considered the *upper bound* of performance without the need to handle undesired demonstrations. The results of the safe policy are presented in Table 6. Overall, UNIQ emerges as the algorithm closest to the Safe policy. In Section 6.4, we further demonstrate that when the amount of undesired data becomes sufficiently large, UNIQ can outperform and achieve a lower cost than a policy trained solely on desired demonstrations.

|  |  | DWBC | SafeDICE | UNIQ | Safe Policy |
|---|---|---|---|---|---|
| Point-Goal-1 | Return | 26.5±0.2 | 26.6±0.2 | 20.7±0.7 | 25.9±0.2 |
|  | Cost | 33.0±3.2 | 36.3±2.9 | **23.5±4.5** | 26.0±2.6 |
| Point-Goal-2 | Return | 26.9±0.1 | 27.0±0.1 | 23.4±0.4 | 25.9±0.2 |
|  | Cost | 45.8±3.4 | 46.8±3.1 | **27.1±3.0** | 26.0±2.6 |
| Point-Goal-3 | Return | 27.1±0.1 | 27.1±0.1 | 26.4±0.2 | 25.9±0.2 |
|  | Cost | 50.2±3.6 | 50.7±3.6 | **40.6±3.1** | 26.0±2.6 |
| Car-Goal-1 | Return | 28.1±1.2 | 29.8±0.8 | 21.0±0.8 | 26.2±0.7 |
|  | Cost | 30.5±3.3 | 36.4±2.9 | **15.4±2.1** | 23.6±2.8 |
| Car-Goal-2 | Return | 32.8±0.7 | 33.5±0.7 | 27.9±0.8 | 26.2±0.7 |
|  | Cost | 47.4±3.8 | 50.5±4.0 | **31.0±2.8** | 23.6±2.8 |
| Car-Goal-3 | Return | 35.0±0.3 | 35.1±0.3 | 34.3±0.4 | 26.2±0.7 |
|  | Cost | 55.3±4.0 | 55.5±3.8 | **53.1±4.1** | 23.6±2.8 |
| Point-Button-1 | Return | 22.0±0.9 | 23.0±0.9 | 8.8±0.7 | 16.4±0.9 |
|  | Cost | 61.5±6.6 | 66.5±6.5 | **12.2±2.7** | 29.1±3.5 |
| Point-Button-2 | Return | 28.3±0.8 | 28.8±0.8 | 10.3±0.9 | 16.4±0.9 |
|  | Cost | 113.2±7.5 | 114.6±8.7 | **19.1±3.0** | 29.1±3.5 |
| Point-Button-3 | Return | 30.4±0.6 | 30.3±0.7 | 14.9±1.1 | 16.4±0.9 |
|  | Cost | 131.5±8.8 | 128.6±8.7 | **55.5±7.6** | 29.1±3.5 |
| Car-Button-1 | Return | 10.1±1.1 | 11.8±1.3 | 2.3±0.4 | 4.4±0.6 |
|  | Cost | 101.0±14.3 | 116.2±13.1 | **35.9±5.4** | 56.7±7.6 |
| Car-Button-2 | Return | 18.7±1.4 | 21.4±1.5 | 5.1±0.7 | 4.4±0.6 |
|  | Cost | 178.0±14.3 | 198.1±15.4 | **65.8±10.1** | 56.7±7.6 |
| Car-Button-3 | Return | 24.1±1.1 | 24.7±1.0 | 14.0±1.5 | 4.4±0.6 |
|  | Cost | 220.9±12.8 | 232.5±10.9 | **144.0±15.2** | 56.7±7.6 |

Table 6: Compare baselines with Safe policy

## C.4 Learning without the Unlabeled Dataset

Since our algorithm is based on Objective 6, which focuses solely on avoiding undesired demonstrations, we want to compare its performance with a version that only avoids these undesired demonstrations:

- IQlearn-UN: Here, IQlearn are only have Undesired dataset. We solve the Objective 6 which only avoid the bad demonstration.
- BC-UN: similar to IQlearn-UN, here, we minimize the log prob of the bad demonstrations.

The results, shown in Figure 7, indicate IQlearn-UN and BC-UN have inconsistent performance. Without support from desired trajectories in the unlabelled dataset, this version fails to learn a correct policy.

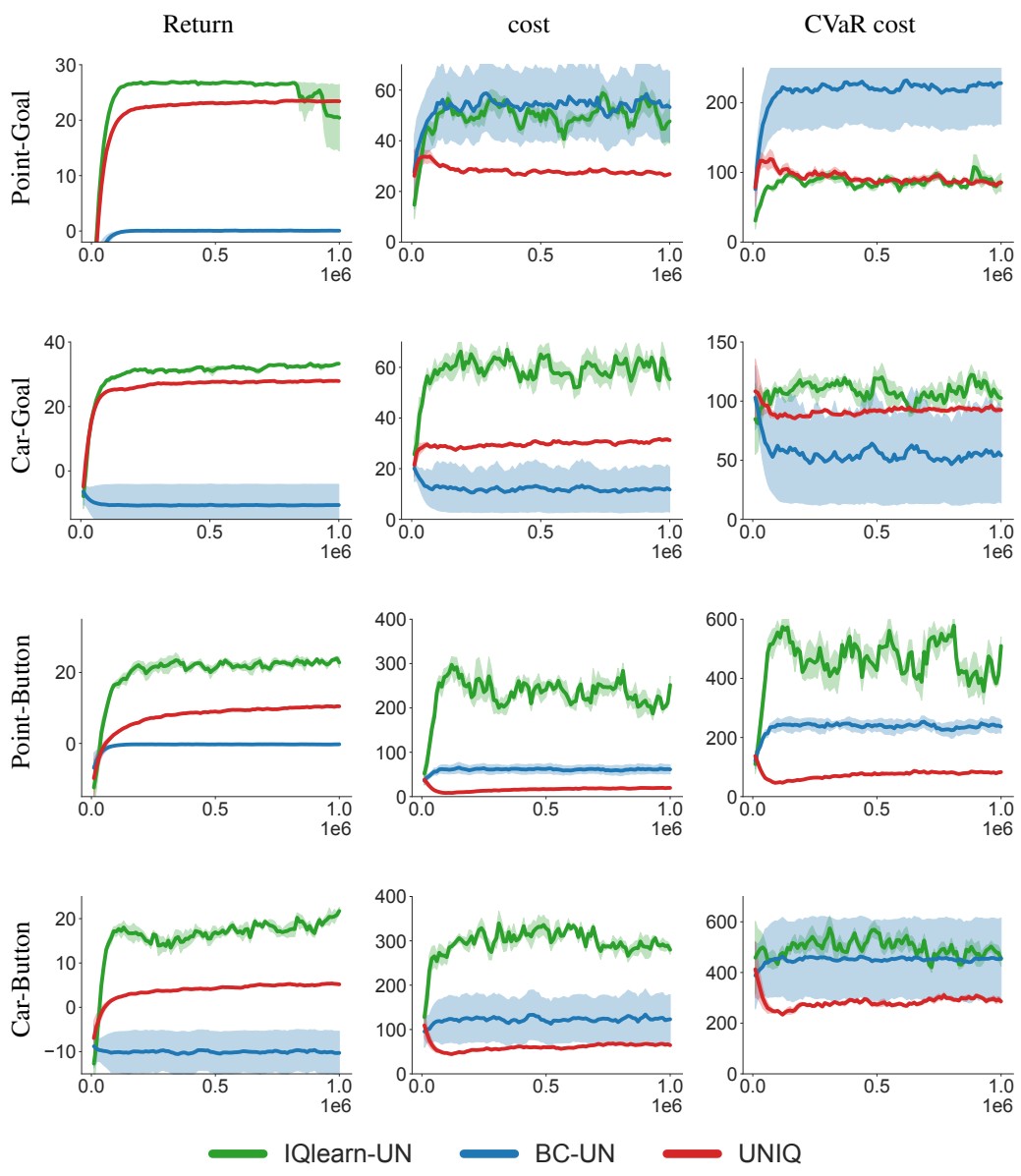

Figure 7: Inconsistent performance in IQlearn-UN and BC-UN compared to the full version of UNIQ.

## C.5 CONTROLLING CONSERVATIVENESS IN UNIQ

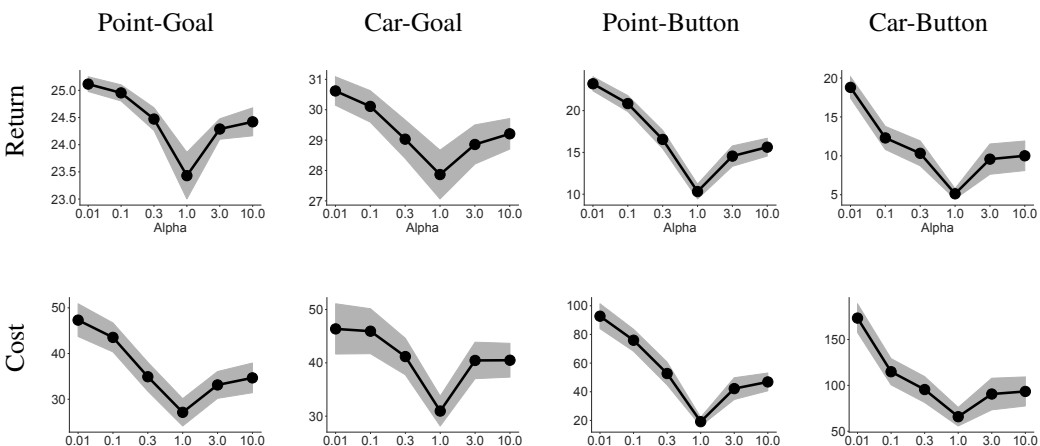

Figure 8: Comparison results of UNIQ with different $\alpha$ selections.

The main paper demonstrates that the policy returned by UNIQ achieves significantly lower costs (indicating safety) but, in some cases, also lower rewards compared to other imitation learning baselines. While this aligns well with our objective of learning safe policies by avoiding unsafe demonstrations, it also raises concerns about the algorithm's conservativeness.

In this section, we show that the conservativeness of UNIQ can be effectively controlled by introducing a parameter to the Weighted BC formulation. Specifically, we adjust the conservativeness of the algorithm by adding a parameter $\alpha$ to the Weighted BC update:

$$\sum_{(s,a)\sim\mathcal{D}^{\mathrm{Mix}}} \exp(\alpha(Q_w(s,a) - V^{Q_w}(s))) \log \pi_\theta(a|s)$$

When $\alpha = 1$, the Weighted BC theoretically returns the exact policy derived from Q-learning, as reported in the main paper. In contrast:

- As $\alpha \to 0$, the Weighted BC returns a random policy.
- As $\alpha \to \infty$, the resulting policy becomes deterministic, always selecting the best action with probability 1.

Thus, by varying $\alpha$, we can deviate the outcome of the Weighted BC from the policy given by Q-learning, reducing the conservativeness of the learned policy.

To experimentally demonstrate this, we vary $\alpha$ and report the corresponding returns and costs on four MuJoCo environments. The results are presented in Figure 8 and Table 7, showing how different values of $\alpha$ impact the trade-off between safety and performance.

Figure 8 demonstrates that UNIQ achieves its safest (and most conservative) performance when $\alpha = 1$. At this value, the policy prioritizes minimizing costs, making it the most risk-averse option. However, as $\alpha$ deviates from 1, both the cost and return increase. This indicates that the Weighted BC formulation produces less conservative policies that are less safe but capable of achieving higher rewards.

Table 7 provides a more detailed breakdown of the costs and returns for different values of $\alpha$. The results show that UNIQ can effectively balance safety and performance: by adjusting $\alpha$, it is possible to achieve a safer policy (i.e., lower cost) while maintaining competitive returns (compared to other baselines). This adaptability highlights the flexibility of UNIQ.

When safety is critical, setting $\alpha = 1$ ensures the most conservative policy, aligning with the objective of avoiding unsafe demonstrations. On the other hand, by varying $\alpha$, one can tune the trade-off to achieve policies that are less safe but yield higher rewards, making UNIQ suitable for a range of

scenarios depending on the desired safety-performance balance. This versatility demonstrates its practicality across different applications with varying safety requirements.

|  | | DWBC | SafeDICE | UNIQ (0.01) | UNIQ (0.1) | UNIQ (0.3) | UNIQ (1.0) | UNIQ (3.0) |
|---|---|---|---|---|---|---|---|---|
| Point-Goal-2 | Return | 26.9±0.1 | 27.0±0.1 | 25.1 ± 0.1 | 25.0 ± 0.1 | 24.5 ± 0.2 | 23.4±0.4 | 24.3 ± 0.2 |
| | Cost | 45.8±3.4 | 46.8±3.1 | 44.0 ± 3.6 | 40.5 ± 3.2 | 32.5 ± 3.2 | **27.1±3.0** | 30.8 ± 2.9 |
| Car-Goal-2 | Return | 32.8±0.7 | 33.5±0.7 | 30.6 ± 0.5 | 30.1 ± 0.5 | 29.0 ± 0.6 | 27.9±0.8 | 28.9 ± 0.6 |
| | Cost | 47.4±3.8 | 50.5±4.0 | 46.4 ± 4.7 | 45.9 ± 4.2 | 41.2 ± 3.4 | **31.0±2.8** | 40.4 ± 3.4 |
| Point-Button-2 | Return | 28.3±0.8 | 28.8±0.8 | 23.2 ± 0.8 | 20.8 ± 1.0 | 16.6 ± 1.1 | 10.3±0.9 | 14.6 ± 1.2 |
| | Cost | 113.2±7.5 | 114.6±8.7 | 92.7 ± 8.7 | 75.9 ± 7.7 | 52.7 ± 8.0 | **19.1±3.0** | 42.2 ± 7.6 |
| Car-Button-2 | Return | 18.7±1.4 | 21.4±1.5 | 18.8 ± 1.4 | 12.3 ± 1.5 | 10.3 ± 1.6 | 5.1±0.7 | 9.6 ± 2.0 |
| | Cost | 178.0±14.3 | 198.1±15.4 | 173.3 ± 15.6 | 114.9 ± 14.3 | 95.5 ± 14.1 | **65.8±10.1** | 90.6 ± 17.3 |

Table 7: Comparison with different $\alpha$

## C.6 EXPERIMENTS ON D4RL WITH RANDOM DEMONSTRATIONS

In this experiment, we test our algorithm UNIQ with unconstrained tasks in D4RL dataset (Fu et al., 2020). We create the unlabeled dataset with the following trajectories:

- 300 expert trajectories.
- 500 medium trajectories.
- 800 random trajectories.

The undesirable dataset consists of 50% medium and 50% random samples. We compare our method using different sizes of the undesirable data. The detailed results are shown in Figure 9. The performance results indicate that in the HalfCheetah-v2 environment, only UNIQ is able to learn effectively. However, in the Ant-v2 environment, DWBC achieves competitive performance compared to UNIQ.

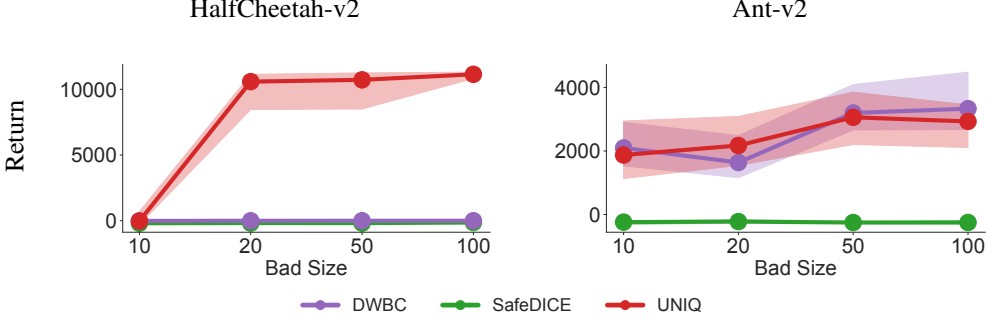

Figure 9: D4RL comparison with increasing size of Undesirable dataset.

## C.7 DISCUSSION ON DSRL

The DSRL dataset (Liu et al., 2024) is a public dataset specifically designed for safe reinforcement learning (RL), generated to encompass every possible pair of return-cost values. This comprehensive coverage helps agents learn to interpret and balance reward and cost signals effectively.

The dataset is constructed using trajectories generated by a mix of safe and unsafe policies, making it particularly challenging for imitation learning methods, including standard approaches and our UNIQ framework, to achieve good performance. The primary challenge lies in the lack of access to reward and cost signals in imitation learning. When expert demonstrations are derived from a variety of policies, current imitation learning methods can only recover a mixture of these policies, rather than isolating the most desirable behaviors. Moreover, since imitation learning does not allow interaction with the environment, the absence of feedback further complicates the learning process. This is in

stark contrast to standard offline RL methods, where reward and cost signals are directly accessible, providing a significant advantage.

To illustrate this difficulty, we used Behavior Cloning (BC), a standard yet effective imitation learning method when sufficient data is available, to learn from a set of expert trajectories in the DSRL dataset. However, BC failed to recover a good policy—the policy it returned deviates significantly from the desirable regions in the return-cost space, as shown in Figure 10. This result highlights the inherent challenges in applying imitation learning methods to DSRL.

We believe that the DSRL dataset is explicitly designed for offline constrained RL and is currently too challenging for existing imitation learning approaches. To our knowledge, no imitation learning methods have been successfully tested on DSRL, and our observations confirm the significant difficulty of applying offline imitation learning in the absence of reward and cost information. These findings reveal a critical limitation of current imitation learning algorithms, including UNIQ, and underscore the need for future research to address these challenges. This dataset presents an opportunity to develop and test new methods capable of overcoming these limitations in offline constrained RL scenarios.

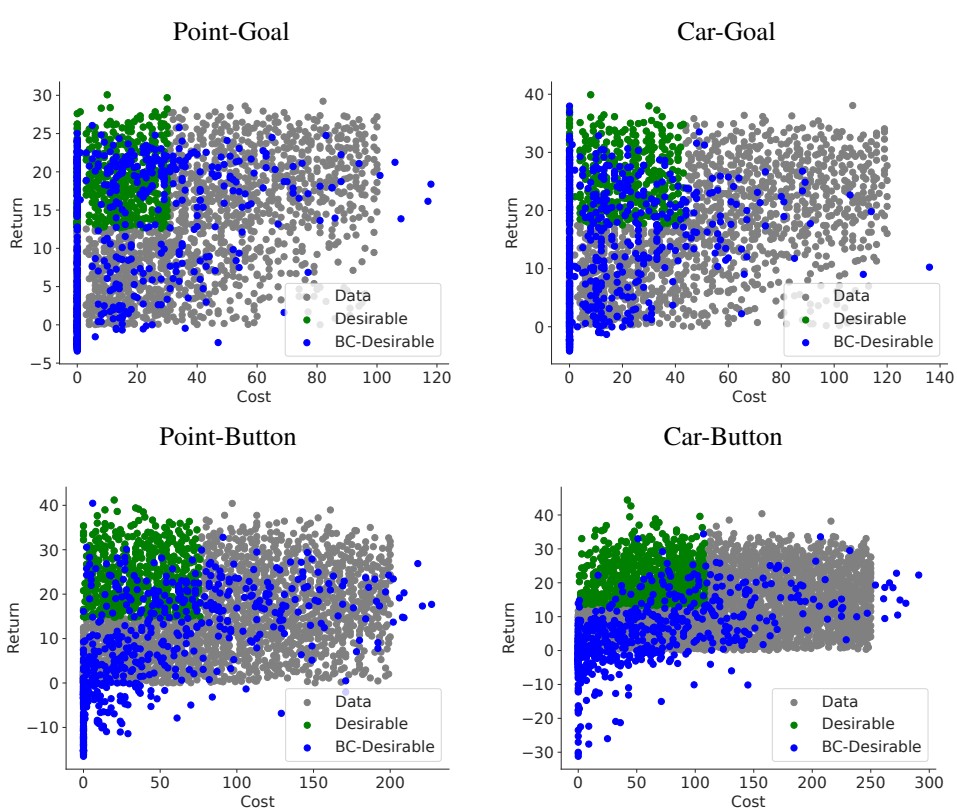

Figure 10: In DSRL dataset, BC from only expert data unable to achieve good performance.

## C.8 FULL NUMERICAL EXPERIMENT FOR MUJOCO VELOCITY TASKS

We evaluate our method on two MuJoCo velocity tasks: Cheetah and Ant. In addition to using a fixed set of 5 trajectories in the undesired dataset, we also test the method with varying sizes of the undesired dataset, annotated as "env-UN= $\{1, 5, 10\}$". The detailed results are summarized in Table 8 and learning curves are shown in Figure 11 and Figure 12. Overall, increasing the size of the undesired dataset helps SafeDICE and DWBC achieve higher performance, while UNIQ reaches its peak performance with just a single undesired trajectory.

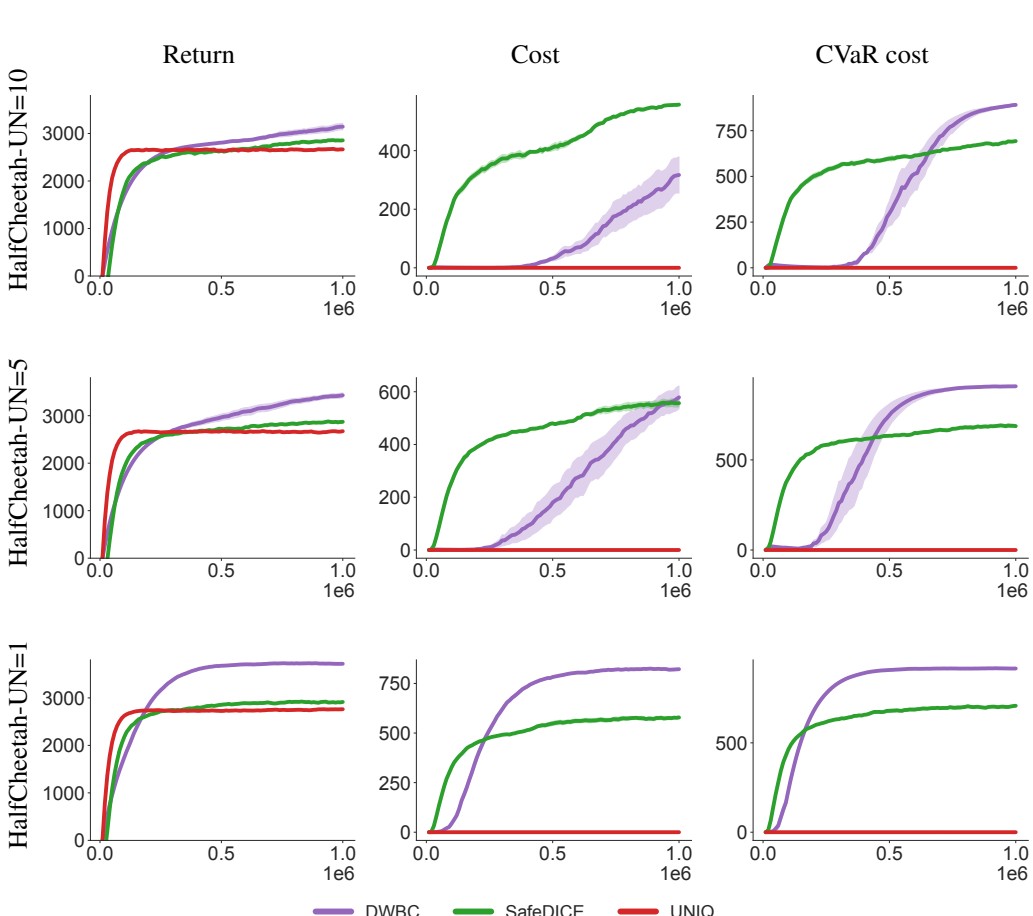

Figure 11: Cheetah task with unlabelled dataset(400-1600) and different undesired dataset.

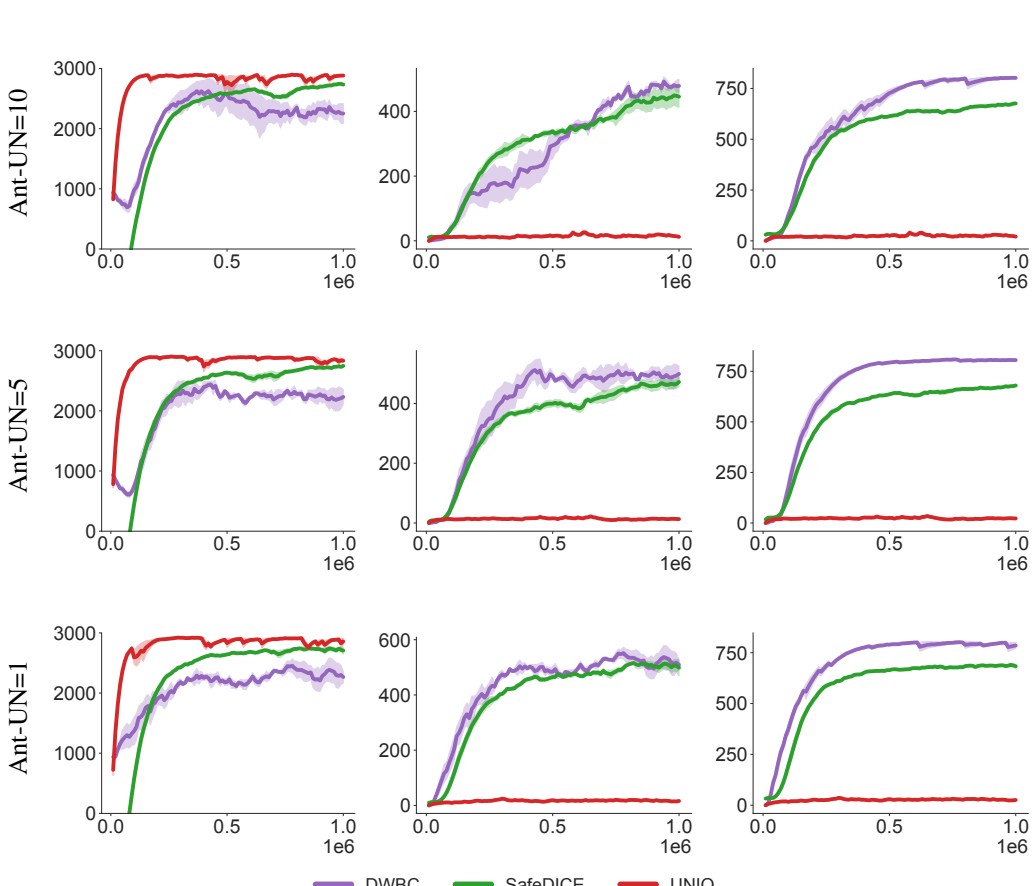

Figure 12: Ant task with unlabelled dataset(400-1600) and different undesired dataset.

|  |  | DWBC | SafeDICE | UNIQ |
|---|---|---|---|---|
| Cheetah-UN=10 | Return | 3135.6±127.4 | 2841.9±56.1 | 2662.0±33.1 |
|  | Cost | 311.0±116.0 | 550.2±13.5 | **0.0±0.0** |
|  | CVaR | 897.7±10.0 | 682.2±14.4 | **0.0±0.0** |
| Cheetah-UN=5 | Return | 3430.9±107.5 | 2860.8±57.8 | 2661.2±29.7 |
|  | Cost | 578.8±89.6 | 553.9±25.3 | **0.0±0.0** |
|  | CVaR | 909.2±6.3 | 686.7±17.3 | **0.0±0.0** |
| Cheetah-UN=1 | Return | 3720.7±39.2 | 2910.0±61.8 | 2755.3±23.8 |
|  | Cost | 823.0±17.5 | 575.5±23.0 | **0.0±0.0** |
|  | CVaR | 916.4±5.0 | 702.2±20.0 | **0.0±0.0** |
| Ant-UN=10 | Return | 2225.0±759.3 | 2713.0±56.2 | 2850.5±177.5 |
|  | Cost | 470.5±162.8 | 439.7±57.7 | **15.2±10.8** |
|  | CVaR | 795.0±103.7 | 668.7±14.6 | **24.6±13.7** |
| Ant-UN=5 | Return | 2210.0±655.7 | 2727.4±49.8 | 2838.2±177.9 |
|  | Cost | 494.5±146.8 | 464.4±35.3 | **13.1±7.5** |
|  | CVaR | 805.7±16.4 | 671.2±16.0 | **22.1±10.5** |
| Ant-UN=1 | Return | 2259.4±653.8 | 2724.4±90.7 | 2841.4±214.9 |
|  | Cost | 507.5±147.8 | 506.5±40.3 | **16.9±7.1** |
|  | CVaR | 789.3±91.3 | 685.8±16.4 | **27.2±9.6** |

Table 8: Full comparison between UNIQ and other baselines in Mujoco-velocity domain. With decreasing of undesirable dataset size, the performance of DWBC and SafeDICE become worse. In contrast, UNIQ able to achieve highest performance with just a single undesirable trajectory.

## C.9 PERFORMANCE WITH THE DATASET EMPLOYED IN THE SAFEDICE PAPER (JANG ET AL., 2024)

As we are using a different dataset from the SafeDICE dataset, we requested the authors of SafeDICE to provide their dataset for comparison. The detailed performance of the expert dataset is shown in Table 9:

|  | Point-Goal | Point-Button |
|---|---|---|
| Mean non-preferred demonstrations cost | 20.018 | 21.933 |
| Mean preferred demonstrations return | 19.911 | 8.286 |
| Mean non-preferred demonstrations cost | 107.977 | 166.099 |
| Mean preferred demonstrations return | 13.798 | 12.085 |

Table 9: SafeDICE dataset performance.

We mix 300 preferred demonstrations and 1200 non-preferred demonstrations for the unlabelled dataset and use 100 non-preferred demonstrations for the undesired dataset. The performance is shown in Figure 13. It is clearly that our method can achieve lower cost than SafeDICE.

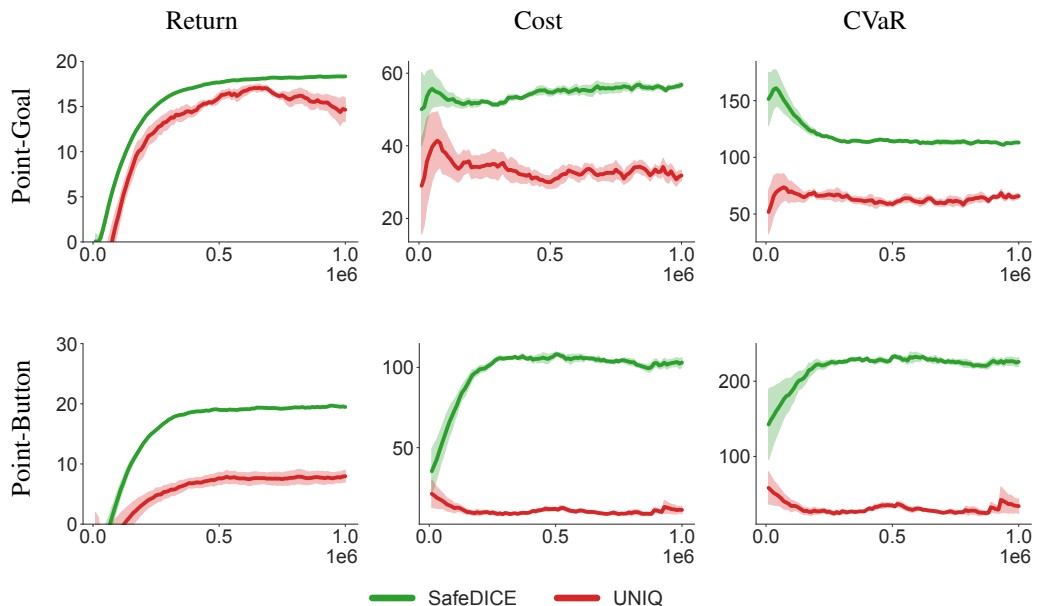

Figure 13: Comparison between UNIQ compared to SafeDICE in their dataset.

## D  DETAILED LEARNING CURVES

### D.1  LEARNING CURVES FOR THE RESULTS REPORTED IN TABLE 1

Learning curve of Table 1 are shown in Figure 14,Figure 15, and Figure 16.

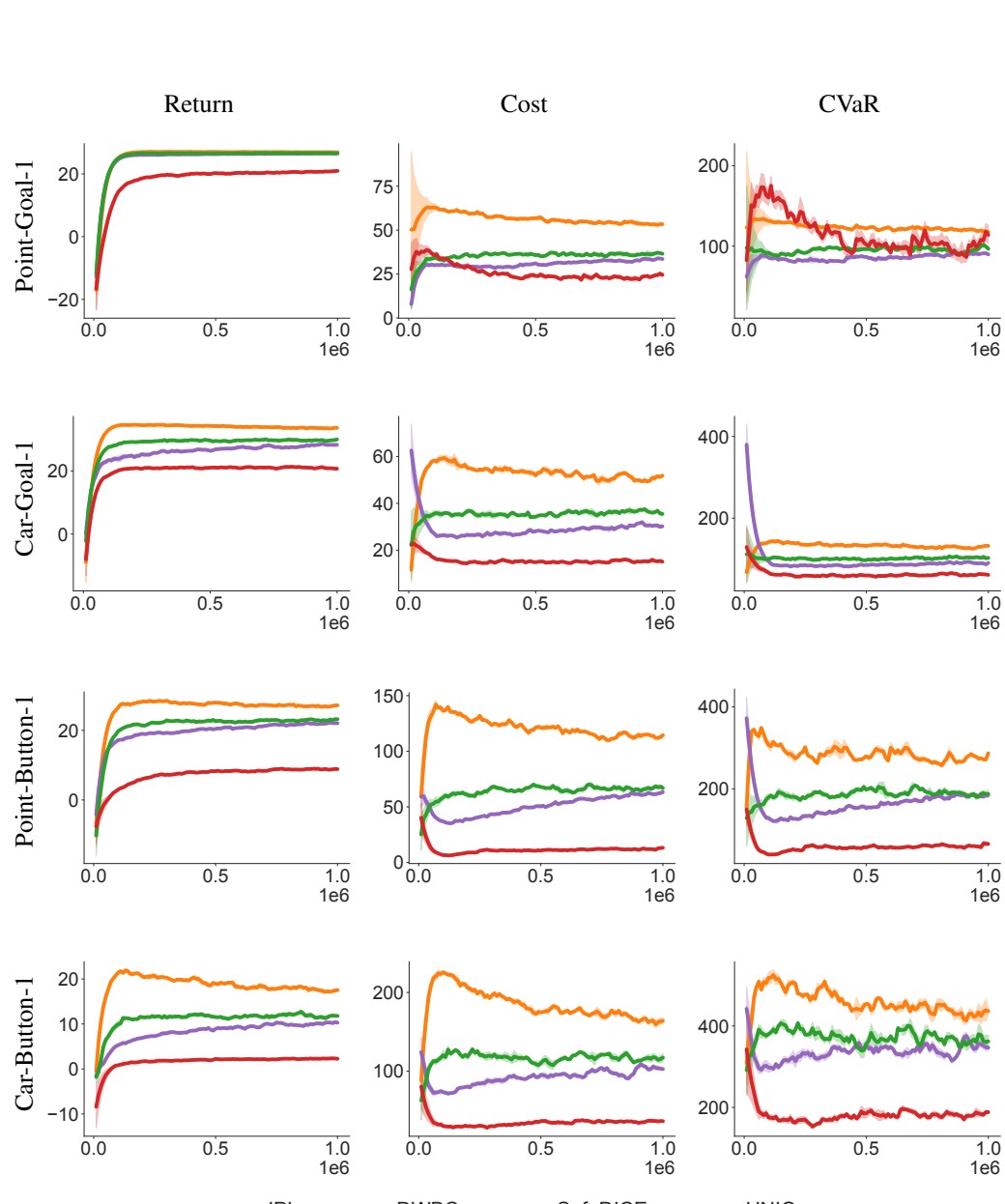

Figure 14: Training curves for difficulty 1 of the Table 1.

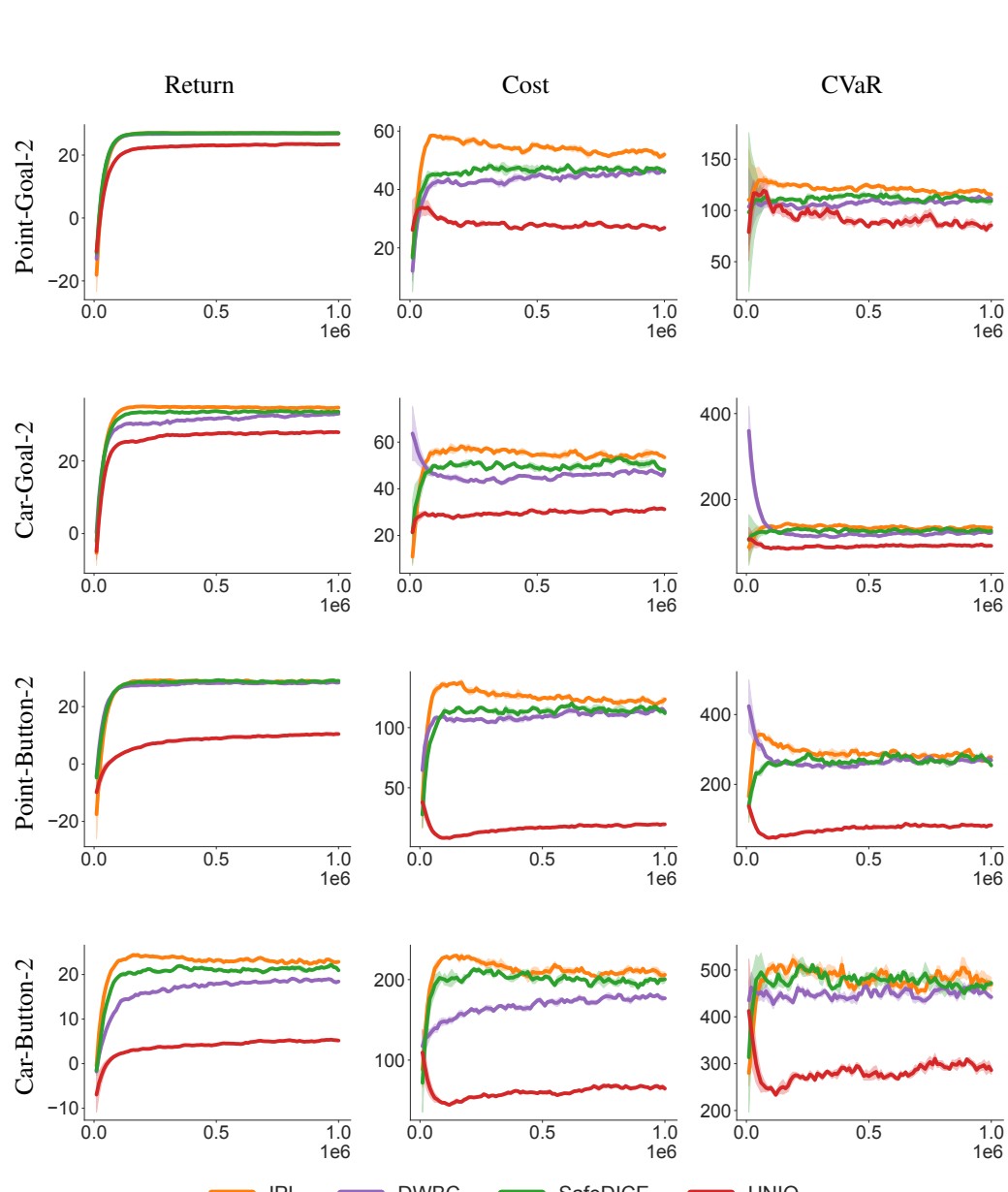

Figure 15: Training curves for difficulty 2 of the Table 1.

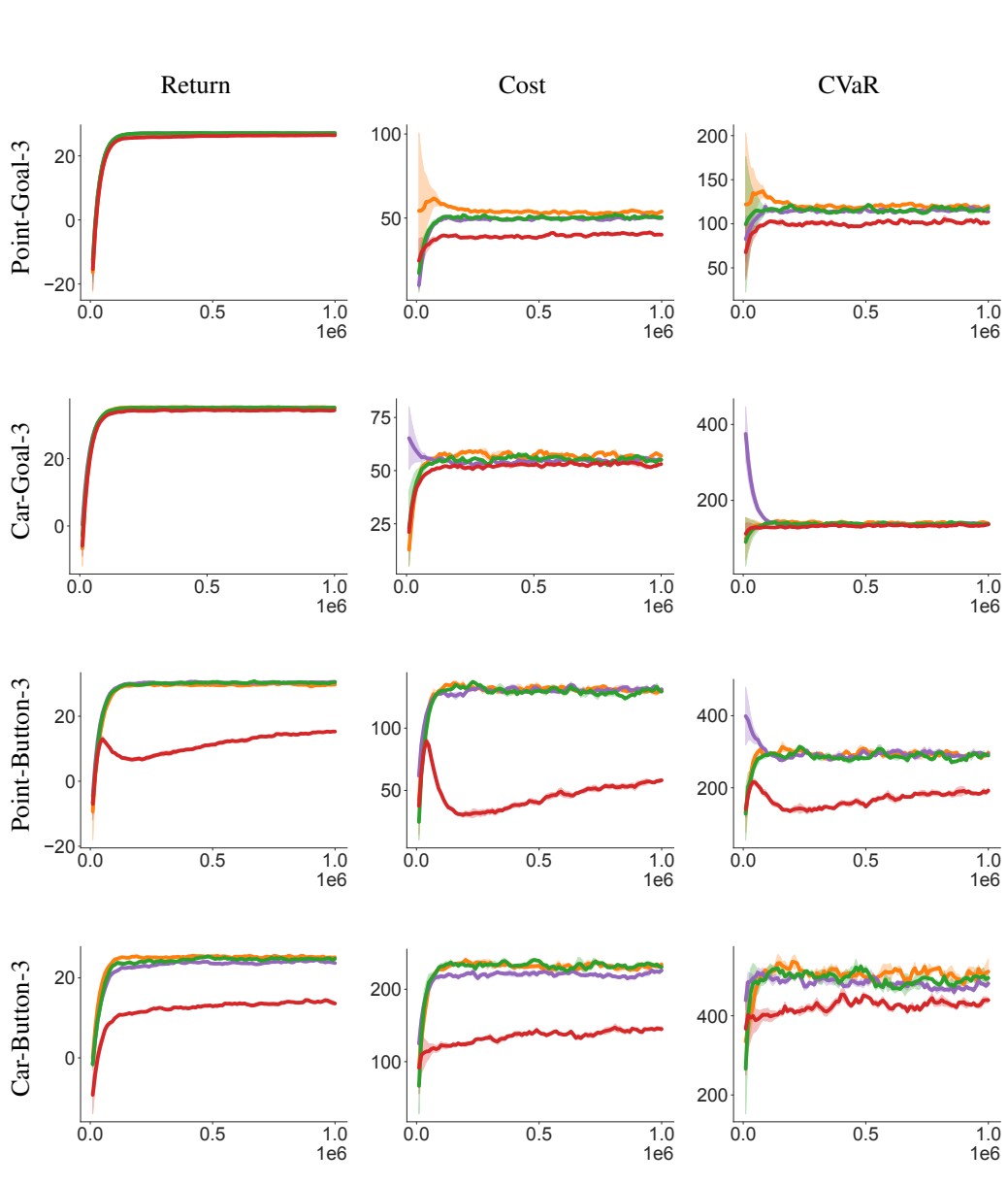

Figure 16: Training curves for difficulty 3 of the Table 1.

## D.2 LEARNING CURVES FOR THE RESULTS REPORTED IN SECTION 6.4

Learning curve of Figure 3. The detailed results are shown in Figure 17 (Point-Goal) and Figure 18 (Car-Goal).

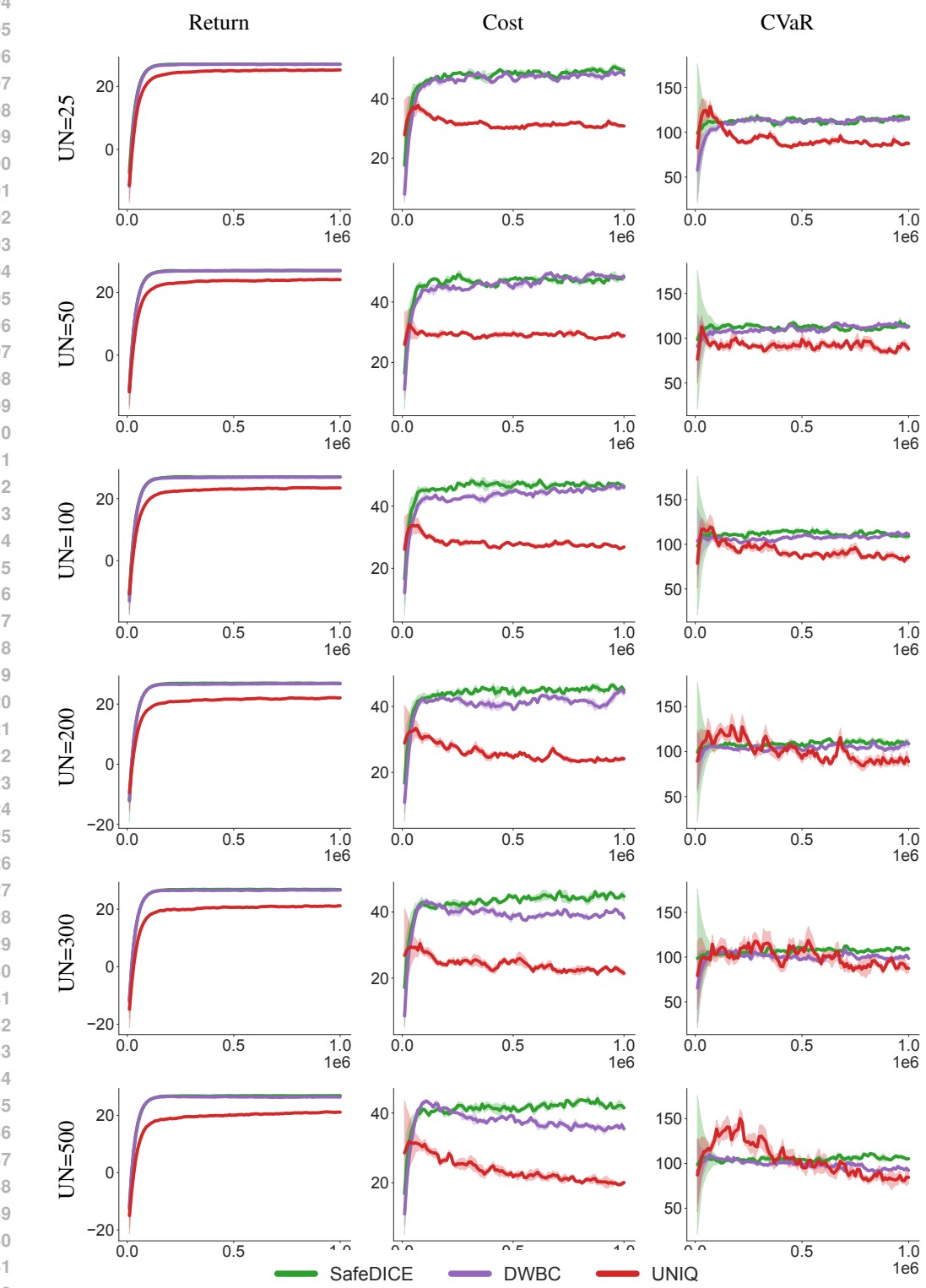

Figure 17: Training curves for Point-Goal task in Figure 3.

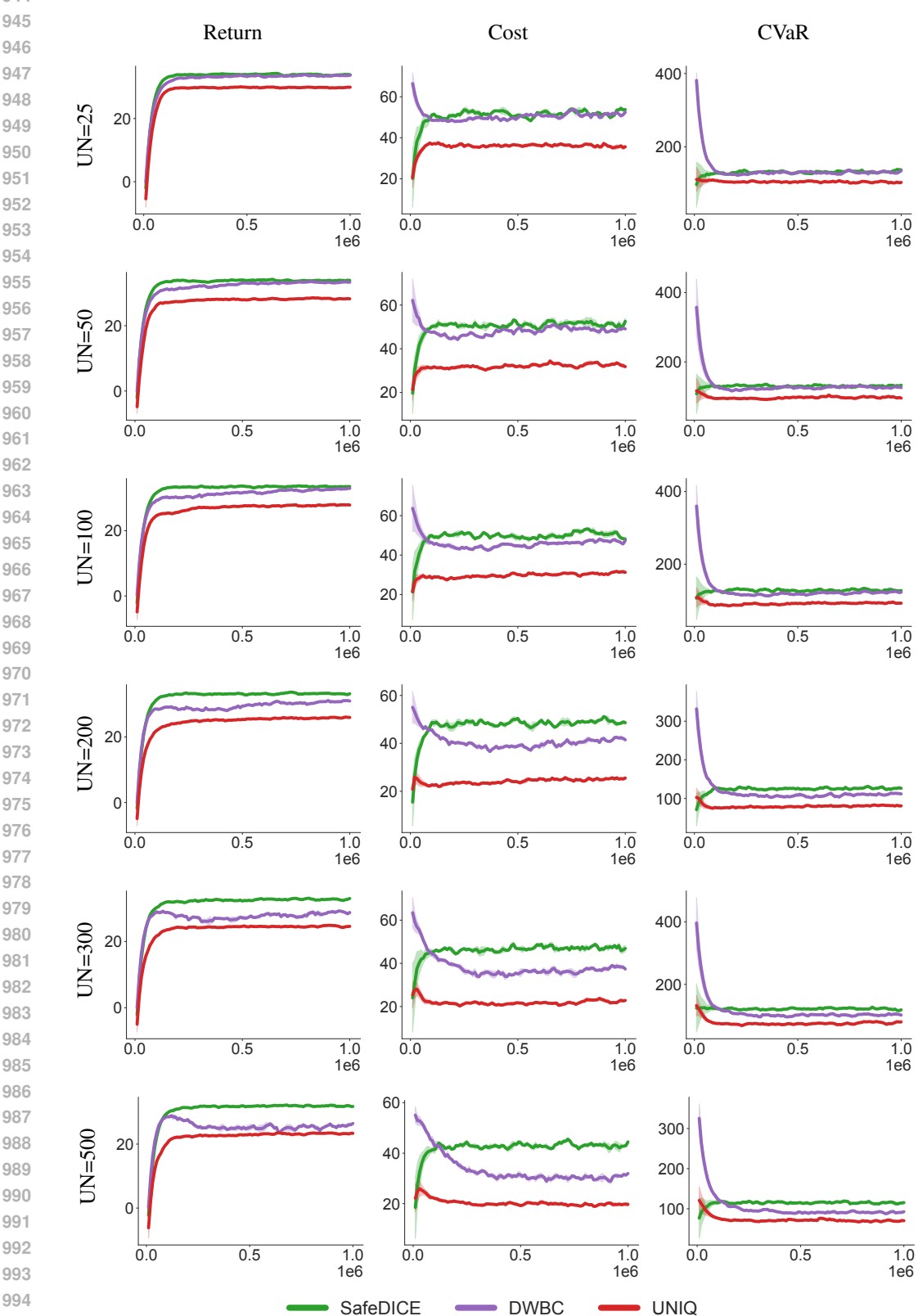

Figure 18: Training curves for Car-Goal task in Figure 3.

