# OpenReview forum: "UNIQ: Offline Inverse  Q-learning for Avoiding Undesirable Demonstrations"
_ICLR.cc/2025/Conference — Submitted to ICLR 2025_

### Official Review · Reviewer_TD6k · 2024-10-31

**Soundness:** 2
**Presentation:** 3
**Contribution:** 2
**Rating:** 3
**Confidence:** 4

**Summary:**

This paper proposes an offline IL method that can leverage both expert or near-optimal demonstrations, as well as a collection of undesirable (sub-optimal) demonstrations. The proposed method aims to imitate the expert behavior while avoiding the undesired behaviors. The proposed method can be seen as a flipped version of IQ-Learn, which instead of minimizing the reward gap, it enlarges the gap with the undesired data distribution. As the IQ-Learn is derived under the online setting, the authors also borrow ingredients from the DICE method to estimate the occupancy ratio as importance weights to enable sampling from the data distribution. Although the problem setting is meaningful, I also have some concerns with the paper. Please see the strengths and weaknesses for detailed comments.

**Strengths:**

- The proposed problem setting is meaningful in practice, which can enable IL policy learning to avoid certain undesirable behaviors.
- Using DICE to convert the IQ-Learn framework to an offline method is interesting.
- The paper is well-organized and easy to read.

**Weaknesses:**

- The proposed method basically flipped the formulation in IQ-Learn, from minimizing the divergence with respect to expert distribution, to maximizing the divergence with respect to undesirable data distribution. Although it sounds reasonable, maximizing a divergence is often not a good idea, as it can produce ill-behaved optimization problems and should be avoided in most algorithm designs. Minimizing a statistical divergence may produce stable optimization towards a unique solution, but maximizing a divergence would encourage optimization in an arbitrary direction. Even under the exact same problem setting, SafeDICE still chooses to minimize some divergence to learn their policy. I have some concerns regarding the stability and solution quality of the proposed method in complex tasks.
- The technical novelty of this paper is incremental, as it is basically an offline, flipped version of IQ-Learn. One new item proposed by the paper is the use of DICE-based method to estimate the occupancy ratio, in order to turn the original IQ-Learn framework from online sampling to offline sampling, but this is still somewhat incremental.
- I also have concerns regarding the heavy dependency on the IQ-Learn framework, as based on my past empirical experience, as well as results reported in many recent IL works, IQ-Learn often performs poorly in many tasks. One reason for this is that the optimization procedure of IQ-Learn is overly complex, which often negatively impacts policy learning stability. The proposed method is even more complex than IQ-Learn (has additional occupancy ratio estimation steps).
- The experiments also have a number of problems.
  - First, the authors use a very strange way to construct the unsafe and safe data (one is collected by an unconstrained PPO policy, and the other is collected with SIM). I don't think this is reasonable as compared to most real-world scenarios. data generated from an RL policy typically have a very narrow data distribution. The datasets generated by the authors would create a data distribution that has two easily separable peaks in distribution, which favors discriminator learning, and also makes the problem easy to solve. The authors should consider evaluating on some more commonly used safe offline RL benchmark datasets, like DSRL[1], which could make the conclusion more convincing.
  - Some baselines are also problematic, which are different from the versions reported in their original paper. The authors need to explain and justify why these changes were made.
    - The original IQ-Learn is derived under the online setting, which is not compatible with the offline setting. In the original IQ-Learn paper, their authors did provide an offline version (see Section 5.1 in the IQ-Learn paper), which changed the way how advantage term is estimated, but this version has some theory-to-practice gaps. The authors mentioned that they use the official implementation of IQ-Learn, but modified the actor update part. I'm not sure if the right offline version of IQ-Learn is compared, since comparing with the online version is meaningless.
    - The "DWBC" baseline is also far from the version reported in the original DWBC paper, with completely different weighting schemes for policy learning.

[1] Liu et al. Datasets and Benchmarks for Offline Safe Reinforcement Learning.

**Questions:**

- Can you provide theoretical or empirical evidence to demonstrate the stability of the proposed approach, particularly in complex tasks?
- How is the performance of the proposed method on DSRL benchmark datasets?
- What are the performances if you use the original offline-version IQ-Learn as well as the original DWBC learning objective as baselines?

---

> ### Author Response · Authors · 2024-11-24
> **We thank the reviewers for the feedback!**
>
> We thank the reviewer for the feedback. Please find you responses to your concerns below.
>
> ----------
>
> > The proposed method basically flipped the formulation in IQ-Learn, from minimizing the divergence with respect to expert distribution, to maximizing the divergence with respect to undesirable data distribution...
>
> We thank the reviewer for the the comments. We agree that maximizing divergence is likely more challenging and less stable than the minimization counterpart. However, our experiments show that the IQ-learn framework, with its reward regularization, can provide a stable training process and yield favorable convergence outcomes.
>
> SafeDICE avoids maximizing divergence by using a standard divergence minimization approach, aiming to mimic a combination of the unlabeled and unsafe occupancies. However, as noted in our paper, this approach has critical issues; when the quality of the unlabeled data is low, the combined policy becomes highly suboptimal for imitation.
>
>  > The technical novelty of this paper is incremental, as it is basically an offline, flipped version of IQ-Learn. One new item proposed by the paper is the use of DICE-based method to estimate the occupancy ratio, in order to turn the original IQ-Learn framework from online sampling to offline sampling, but this is still somewhat incremental.
>
> Thank you for the comment. We would like to highlight that our formulation differs significantly from many other maximum entropy-based algorithms in the literature. The reversed approach considered in our paper breaks several properties of the standard MaxEnt framework, necessitating novel explorations.
>
> > I also have concerns regarding the heavy dependency on the IQ-Learn framework, as based on my past empirical experience, as well as results reported in many recent IL works, IQ-Learn often performs poorly in many tasks...
>
> Thank you for the comment. To our knowledge, IQ-learn framework remains the state-of-the-art approach for both online and offline imitation learning, whether learning from expert or sub-optimal demonstrations [1]. Moreover, IQ-learn (and the reversed version proposed in our paper) are relatively straightforward to implement. Our experimental results demonstrate that it performs stably and outperforms several baseline methods (e.g., BC, DWBC, DICE-based algorithms). Additionally, our algorithm is no more complex to implement than standard IQ-learn.
>
> > The experiments also have a number of problems. First, the authors use a very strange way to construct the unsafe and safe data ...
>
> We thank the reviewer for the insightful comments. In our experiments, we generated unsafe data by running unconstrained PPO, as it typically produces unsafe trajectories (high cost, albeit with high reward). For safe demonstrations, we used SIM, a state-of-the-art approach in safe RL, which guarantees trajectories with low cost. We believe this setup aligns well with our objective of learning to avoid unsafe demonstrations.
>
> Your comment about the DSRL dataset is well taken, and we agree that using data from various RL policies would be more realistic and challenging. We explored the DSRL dataset and found that, while it is designed for testing constrained RL algorithms, it is exceedingly challenging for imitation learning algorithms where reward and cost information is inaccessible. We have added a detailed discussion in Section C7 of the appendix (page 27 in the revised draft), highlighting that under such data configurations, even standard “learning-from-expert” demonstration approaches face significant difficulties. For instance, we tested Behavior Cloning (BC) with a large number of good (safe) demonstrations, but it still failed to learn an effective policy.
>
> We believe that no imitation learning methods have yet been successfully tested on DSRL, and our observations confirm the substantial challenges of applying offline imitation learning in the absence of reward and cost signals. These findings reveal a critical limitation of current imitation learning algorithms, including UNIQ, and emphasize the need for future research to address these challenges. The DSRL dataset presents an excellent opportunity to develop and test new methods capable of overcoming these limitations in offline constrained RL scenarios.
>
> We have acknowledged this as a limitation of our current work (in Section C7, page 27 in the revised draft) and will certainly explore it further in future research.
>
> We have conducted additional experiments on D4RL, another widely-used public dataset, to further evaluate our approach. The results demonstrate that our method performs effectively in learning from bad demonstrations (i.e., demonstrations with low rewards). For details, please refer to Section C6 (page 27 in the revised draft).
>
> [1] Dual RL: Unification and New Methods for Reinforcement and Imitation Learning" Harshit Sikchi, Qinqing Zheng, Amy Zhang, and Scott , ICLR2024

---

> ### Author Response · Authors · 2024-11-24
>
> > Some baselines are also problematic, which are different from the versions reported in their original paper. The authors need to explain and justify why these changes were made. The original IQ-Learn is derived under the online setting, which is not compatible with the offline setting...
>
> We thank the reviewer for the insightful comment. It is true that our implementation is based on the official version of IQ-Learn, and we must address the challenge of the offline setting where we only have an unlabeled dataset instead of online interaction. However, some prior works [1,2] have addressed this problem, and our implementation, incorporating techniques from these papers, successfully works in the offline setting.
>
> > The "DWBC" baseline is also far from the version reported in the original DWBC paper, with completely different weighting schemes for policy learning.
>
> We thank the reviewer for the comment. The original DWBC is not designed for learning from undesirable demonstrations, so we have made our best effort to adapt it to work within our context for the sake of comparison.
>
> > Can you provide theoretical or empirical evidence to demonstrate the stability of the proposed approach, particularly in complex tasks?
>
> We believe our experiments demonstrate that UNIQ is stable across most standard benchmarking tasks, consistently outperforming other baselines. Additionally, in the updated paper, we have included new results that illustrate how conservativeness within UNIQ can be effectively controlled, as well as experiments using the public D4RL dataset (section C6 in appendix, page 27 in the revised draft). We hope these additional results and explanations adequately demonstrate the stability and robustness of our approach.
>
> > What are the performances if you use the original offline-version IQ-Learn as well as the original DWBC learning objective as baselines?
>
> The original IQ-Learn and DWBC are designed to mimic expert or near-expert demonstrations, which contrasts with our context, where only undesirable demonstrations are available. Directly applying IQ-Learn and DWBC in this setting would be inappropriate, as the goal is to avoid mimicking undesirable behaviors. Therefore, we believe the adapted versions used in our paper represent the most direct and reasonable approach to address the unique challenges of our setting.
>
>
> [1] "Dual RL: Unification and New Methods for Reinforcement and Imitation Learning" Harshit Sikchi, Qinqing Zheng, Amy Zhang, and Scott , ICLR 2024
>
> [2] "SPRINQL: Sub-optimal Demonstrations driven Offline Imitation Learning" Hoang, Huy, Tien Anh Mai, and Pradeep Varakantham, NeurIPS 2024.
>
> ----------
>
> **We hope our revisions and our response  address the reviewers’ concerns and further clarify our contributions. If there are any additional questions or comments, we would be happy to address them.**

---

> > ### Comment · Reviewer_TD6k · 2024-11-26
> > **Thanks for the response**
> >
> > Thank you for the detailed response, but many of my previous concerns still remain. Specifically,
> > - Regarding the response to divergence maximization: I don't think the argument from the authors is strong enough. The problem itself is ill-behaved (two other reviewers also share the same concern), simply arguing based on observations from limited experiments is not very convincing.
> > - Regarding the novelty: how can you claim your method "differs significantly" from other methods, if the majority of your algorithm is a flipped version (from min to max) of another well-known algorithm? People do not adopt the reversed approach previously primarily because it is mathematically problematic/
> > - Regarding the performance of IQ-Learn: I don't think IQ-Learn can be called a SOTA method. There are quite many papers showing IQ-Learn has a bad performance. For example, if the authors check the DualRL paper mentioned in your response, it also has IQ-Learn as a baseline, and the performance is quite poor. This is consistent with my past experience with running IQ-Learn, it's kind of hard to get reasonable performance as compared to more recent offline IL methods.
> > - Regarding the test dataset: if the authors check the offline safe IL extension of [1] (see its Section 4.1), it verifies safe IL performance on the DSRL benchmark. Moreover, many previous offline IL with mixed-quality dataset methods verify their methods using the composition of the D4RL datasets (e.g., DWBC and DualRL). I still can't see why it is not possible to validate on the standard benchmark datasets.
> >
> > [1] Safe Offline Reinforcement Learning with Feasibility-Guided Diffusion Model. ICLR 2024.
> >
> > - Regarding DWBC baseline: yes, the problem setting is different, but you only need to change the positive-unlabeled learning objective to the negative-unlabeled learning objective, this will result in changing the sign of some parts of its BC weights. However, the version of "DWBC" used in this paper used an extremely naive weighting scheme, which is far from the one used in DWBC.

---

### Official Review · Reviewer_49zx · 2024-11-05

**Soundness:** 2
**Presentation:** 2
**Contribution:** 2
**Rating:** 3
**Confidence:** 4

**Summary:**

The authors present a new framework for training a desired policy using two datasets: one containing undesirable trajectories and another with unlabeled data. The goal is to train the policy without an explicit definition of optimal behavior, relying instead on a concept of behaviors to avoid. The authors argue that this approach is crucial in fields like autonomous driving and healthcare, where unsafe actions could lead to catastrophic outcomes.

**Strengths:**

- the concept is interesting and could indeed have important applications
- minor modification to existing approach, making application more straightforward
- the paper is well written

**Weaknesses:**

- By merely avoiding certain behaviors, the resulting solution space appears vast and highly ambiguous.
- It seems that the undesired dataset needs to be quite large to effectively narrow the space of desired behaviors. The experiments suggest that a substantial number of trajectories were used.
- The experiments also indicate that the unlabeled dataset may actually contain safe trajectories rather than genuinely unlabeled data, implying a notion of optimality within the datasets. These safe trajectories could be viewed as an expert distribution to match, making them suitable for standard imitation learning methods. (see questions)
- I would argue that the datasets used in this paper are not truly unlabeled but instead consist of trained behaviors—one representing safe behavior and the other unsafe—leading to a narrow solution space. I doubt that this approach would scale effectively to settings with genuinely unlabeled data. It would be valuable for the authors to include an ablation study addressing this point. They could, for instance, test scalability by using a small set of undesired behaviors along with some of the untrained D4RL dataset to validate their approach.

**Questions:**

- What is the divergence optimized here?
    -  [1] demonstrated that IQ-Learn minimizes the chi-squared divergence between the policy and a mixture distribution. Here, however, it seems that the approach maximizes the chi-squared divergence between a mixture distribution and another mixture distribution that includes both desired and undesired behaviors.
- How were the baselines trained, like BC-Mix and IQ-learn Mix? Where they trained to imitate only the Mix distribution? If yes, I would argue that this is an unfair since you actually have access to the dataset you would like to imitate (the safe trajectories), and IQ-learn and BC should rather try to imitate that one.

[1] Al-Hafez et. al, LS-IQ: Implicit Reward Regularization for Inverse Reinforcement Learning

Minor Points:
- try to write more compactly:
    - huge tables should be in appendix, only a reduced version should be in the main paper
- many of the propositions are very trivial extension previous ones, and probably not worth to specifically include them in the actual paper.

---

> ### Author Response · Authors · 2024-11-24
> **We thank the reviewers for the feedback!**
>
> We thank the reviewer for the comments. Please find below our responses to your concerns.
>
> ------
>
> > By merely avoiding certain behaviors, the resulting solution space appears vast and highly ambiguous.
>
> We thank the reviewer for the insightful comments.
>
> We agree with the reviewer that when avoiding bad behavior, the solution space becomes large and more ambiguous compared to the standard setting of learning from expert behavior. This, however, should not be viewed as a limitation of our approach—since there are no expert demonstrations in our setting, our objective is not to learn an optimal or near-optimal policy but rather to avoid undesirable behavior as much as possible.
>
> > It seems that the undesired dataset needs to be quite large to effectively narrow the space of desired behaviors. The experiments suggest that a substantial number of trajectories were used.
>
> It is true that, in our context, UNIQ requires a larger number of undesirable demonstrations to effectively narrow the learned policy to the desirable region. We believe this is both reasonable — learning effectively from mistakes necessitates a broader set of demonstrations to ensure the avoidance of potential errors. Conversely, when good demonstrations are available, the learning process can simply focus on following these desirable examples, requiring fewer instances.
>
> > The experiments also indicate that the unlabeled dataset may actually contain safe trajectories rather than genuinely unlabeled data, implying a notion of optimality within the datasets. These safe trajectories could be ...
>
> While the mixed demonstrations include expert trajectories, identifying them is challenging—and even impossible—due to the lack of reward and cost information. Furthermore, the mixed dataset contains undesirable trajectories, which means that applying standard imitation learning methods would result in mimicking these undesirable behaviors, leading to suboptimal and potentially unsafe policies.
>
> In our experiments, we have demonstrated that standard imitation learning methods applied to the mixed dataset do not perform well, further highlighting the limitations of such approaches in this context.
>
> > I would argue that the datasets used in this paper are not truly unlabeled but instead consist of trained behaviors—one representing safe behavior and the other unsafe—leading to a narrow solution space. I doubt that this approach would scale effectively to settings with genuinely unlabeled data...
>
> Thank you for the suggestion regarding the D4RL dataset. In response, we have added experiments using the D4RL dataset to Section C6 of the appendix in the updated paper (page 27). In these experiments, we treat the random (or untrained) dataset as undesirable. The results demonstrate that UNIQ is effective in learning from this dataset, further validating its robustness.
>
> > What is the divergence optimized here?  [1] demonstrated that IQ-Learn minimizes the chi-squared divergence between the policy and a mixture distribution. Here, however, it seems that the approach maximizes the chi-squared divergence between a mixture distribution and another mixture distribution that includes both desired and undesired behaviors.
>
> > [1] Al-Hafez et. al, LS-IQ: Implicit Reward Regularization for Inverse Reinforcement Learning
>
> Thank you for the comment. In our context, the divergence between the undesirable and learning policy is maximized (as shown in our Proposition 1).
>
> > How were the baselines trained, like BC-Mix and IQ-learn Mix? Where they trained to imitate only the Mix distribution? If yes, I would argue that this is an unfair since...
>
> Thank you for the comments. To clarify, we do not have access to safe trajectories. In our setting, we only have undesirable and mixed demonstrations, with the objective of learning a good policy from this data. Since BC and IQ-learn are specifically designed to mimic the data, we believe that applying them to the mixed demonstrations (which are at least not entirely undesirable) is the best adaptation for our setting.

---

> ### Author Response · Authors · 2024-11-24
>
> > Minor Points:
>
> > try to write more compactly:
> huge tables should be in appendix, only a reduced version should be in the main paper.
>
> Thank you for the suggestion. We will update the tables to make them more compact.
>
> > many of the propositions are very trivial extension previous ones, and probably not worth to specifically include them in the actual paper.
>
> We thank the reviewer for the suggestion. We would like to emphasize that our objective function differs significantly from those in previous works such as MaxEnt or IQ-Learn. Specifically, in IQ-Learn, the objective is to maximize the reward of expert transitions while minimizing others, whereas in our case, the objective is to minimize undesirable demonstrations while maximizing others. This fundamental difference means that the theoretical findings from previous works do not directly apply, and our new objective requires a substantially novel exploration.
>
> We believe that Propositions 4.1 and 4.2 are non-trivial contributions. In fact, they are quite surprising, as they demonstrate that despite changing the objective to maximize a divergence, some important properties of previous IQ-Learn methods can still be retained. This insight underscores the novelty and significance of our theoretical findings.
>
> ---------
>
> **We hope our response  address the reviewers’ concerns and further clarify our contributions. If there are any additional questions or comments, we would be happy to address them.**

---

> > ### Comment · Reviewer_49zx · 2024-11-30
> >
> > Dear authors,
> >
> > thank you for the detailed response. I still have some remaining points.
> >
> > > > The experiments also indicate that the unlabeled dataset may actually contain safe trajectories rather than genuinely unlabeled data, implying a notion of optimality within the datasets. These safe trajectories could be ...
> >
> > > While the mixed demonstrations include expert trajectories, identifying them is challenging—and even impossible—due to the lack of reward and cost information. Furthermore, the mixed dataset contains undesirable trajectories, which means that applying standard imitation learning methods would result in mimicking these undesirable behaviors, leading to suboptimal and potentially unsafe policies.
> >
> > > In our experiments, we have demonstrated that standard imitation learning methods applied to the mixed dataset do not perform well, further highlighting the limitations of such approaches in this context.
> >
> > I do not fully agree here. I would argue that it is possible to identify them, since non-expert demonstrations are in the undesired dataset. And optimizing under mixture distributions is nothing uncommon, many approaches like LS-IQ, Dual-RL, or IQ-Learn at least in practice do so.
> >
> > > > I would argue that the datasets used in this paper are not truly unlabeled but instead consist of trained behaviors—one representing safe behavior and the other unsafe—leading to a narrow solution space. I doubt that this approach would scale effectively to settings with genuinely unlabeled data...
> >
> > > Thank you for the suggestion regarding the D4RL dataset. In response, we have added experiments using the D4RL dataset to Section C6 of the appendix in the updated paper (page 27). In these experiments, we treat the random (or untrained) dataset as undesirable. The results demonstrate that UNIQ is effective in learning from this dataset, further validating its robustness.
> >
> > I was specifically saying that the authors should try to train their algorithm without expert demonstrations. In the added D4RL example, the mixed dataset again contains a significant amount of expert demonstrations. I would like to see an example without expert demonstration, but rather only medium.
> >
> > > What is the divergence optimized here? [1] demonstrated that IQ-Learn minimizes the chi-squared divergence between the policy and a mixture distribution. Here, however, it seems that the approach maximizes the chi-squared divergence between a mixture distribution and another mixture distribution that includes both desired and undesired behaviors.
> >
> > > [1] Al-Hafez et. al, LS-IQ: Implicit Reward Regularization for Inverse Reinforcement Learning
> >
> > > Thank you for the comment. In our context, the divergence between the undesirable and learning policy is maximized (as shown in our Proposition 1).
> >
> > I was rather asking for the specific kind of divergence. Maybe the authors have some intuition here.
> >
> > At least two other reviewers seem to have similar concerns on related topics.

---

### Official Review · Reviewer_AdDM · 2024-11-08

**Soundness:** 2
**Presentation:** 3
**Contribution:** 2
**Rating:** 3
**Confidence:** 4

**Summary:**

This work tackles the challenge of offline learning in scenarios with undesirable demonstrations, aiming to train agents that avoid unsafe or undesired behaviors in environments where undesirable and mixed demonstrations are given.
The paper presents UNIQ (**UN**desirable demonstrations-driven **I**nverse **Q**-Learning), an approach built on an inverse Q-learning (IQ-Learn) framework.
By exploiting IQ-Learn framework, the study aims to learn a reward function that penalizes actions similar to those in the undesirable dataset, effectively steering the policy away from risky behaviors.
Instead of directly optimizing within the policy and reward spaces, UNIQ operates in the Q-space, following a similar approach to IQ-Learn.
The optimized Q-function is then used to derive the policy through weighted behavior cloning, with weights informed by an advantage function computed from the Q-values.
Through this approach, UNIQ efficiently learns a policy that emphasizes safety by penalizing actions aligned with undesirable demonstrations, making it a robust method for offline learning in risk-sensitive environments.

**Strengths:**

This work integrates the IQ-Learn framework into the problem setting of learning from undesirable demonstrations, addressing a challenge in safe offline imitation learning.
By extending IQ-Learn to handle scenarios with undesired behaviors, the authors provide a novel application that moves beyond traditional offline imitation learning and into the realm of safe policy learning, which is important in real-world applications.

The study goes beyond merely applying IQ-Learn to this setting; it rigorously justifies the suitability of IQ-Learn for learning from undesirable demonstrations.
Through in-depth analysis, the authors clarify how the proposed objective captures the characteristics needed to discourage unsafe behaviors, and this thorough justifications strengthen the reliability for applications of UNIQ.

Additionally, the authors propose a practical version of the UNIQ algorithm, making it applicable to real-world scenarios where undesirable demonstrations are given.
This implementation, combined with theoretical insights, enables safe policy learning in some practical offline learning scenarios.

**Weaknesses:**

**1. Questionable Design of Training Objective**

The training objective (Eq. 3) of UNIQ is focused solely on maximizing the discrepancy between the state-action distributions of non-preferred and imitator demonstrations.
However, this objective lacks a mechanism to guide the policy toward with expert behaviors contained in mixed demonstration datasets.
Consequently, the Q-function is primarily trained to penalize undesired demonstrations rather than to increase the Q-value of expert demonstrations.
This implies that when the unlabeled dataset contains a large number of safe yet non-optimal random actions in $D^{MIX} \setminus D^{UN}$, UNIQ may struggle to identify safe expert behaviors.

While this may result in a "safe" agent but it raises concerns regarding its usefulness for real-world applications.
For instance, in an autonomous driving context, a policy that avoids non-preferred behaviors by simply stopping at the starting state could technically be safe,  yet it would lack the functional behaviors that essential for practical deployment.

**2. Limited Consideration of Experimental Scenarios**

The experimental results also highlights a limitation of the current objective design, as UNIQ achieves relatively low costs and returns, indicating safety but at the expense of effectiveness.
These results appear insufficient to demonstrate that UNIQ is practically beneficial, as all evaluations are conducted on constrained RL benchmarks that primarily aim to maximize returns while keeping costs within predefined thresholds.

As discussed in Section 1, the authors may argue that UNIQ is advantageous in certain scenarios — such as when expert demonstrations are entirely absent in the mixed dataset or when specific undesirable actions must be avoided by pretrained agents without compromising performance — the absence of experiments on these contexts weakens the argument for its applicability.
The authors are encouraged to explore and present alternative scenarios and corresponding experiments that are suitable for applying the method's objectives and this would strengthen the justification of UNIQ’s approach and its practical relevance.

**Questions:**

Q1. In line 220, the manuscript assumes that “$D^{MIX}$ may contain a mix of random, undesired, and expert demonstrations.” However, datasets used in experiments comprised with undesired and expert demonstrations. Could you provide additional exprimental results with mixing (safe) random demonstrations to the dataset?

Q2. Could you clarify what is meant by “the quality of the unlabeled dataset” in line 244? Does this refer to the proportion of preferred and non-preferred demonstrations in $D^{MIX}$? Additionally, why does UNIQ rely less on this compared to SafeDICE?

**Details Of Ethics Concerns:**

I have no ethical concern on this work.

---

> ### Author Response · Authors · 2024-11-24
> **We thank the reviewers for the feedback!**
>
> We thank the reviewer for the comments. Please find below our responses to your concerns.
>
> -------------
> > 1. Questionable Design of Training Objective. The training objective (Eq. 3) of UNIQ is focused solely on maximizing the discrepancy between the state-action distributions of non-preferred and imitator demonstrations. However, this objective lacks a mechanism to guide the policy toward with expert behaviors...
>
> We thank the reviewer for the insightful comments. We agree that UNIQ does not explicitly guide the policy toward expert behavior in a mixed dataset. However, this should not be considered a limitation of UNIQ, as our framework is focused on prioritizing safety. More specifically, we make no assumptions about the nature of the mixed dataset—it may include undesirable behaviors. In such cases, relying on an approach that mimics expert behavior from the mixed dataset could lead to unsafe outcomes.
>
> In scenarios where it is known that the mixed dataset contains mostly reasonable demonstrations, one might consider alternative algorithms designed for learning from suboptimal demonstrations. However, our algorithm and problem formulation are specifically tailored to settings where undesirable demonstrations must be explicitly avoided (e.g., unsafe actions in autonomous driving or healthcare). This focus ensures that the policy learns to prioritize safety while effectively handling the presence of undesirable behaviors in the dataset.
>
> > While this may result in a "safe" agent but it raises concerns regarding its usefulness for real-world applications. For instance, in an autonomous driving context, a policy that avoids non-preferred behaviors by simply stopping at the starting state could technically be safe, yet it would lack the functional behaviors that essential for practical deployment.
>
> We would like to note that our primary goal is to avoid undesirable demonstrations. Undesirable does not mean only unsafe trajectories, it can also be low reward but safe trajectories (e.g., stopping in the middle of the road to avoid collisions).  In the context of safe RL, undesirable demonstrations may include unsafe trajectories or safe but low-reward trajectories. For the example you mentioned, the stopping demonstration has a low reward and could be marked as undesirable, so it should be avoided.
>
> > 2. Limited Consideration of Experimental Scenarios: The experimental results also highlights a limitation of the current objective design, as UNIQ achieves relatively low costs and returns, indicating safety but at the expense of effectiveness. These results appear insufficient to demonstrate that UNIQ is practically beneficial, as all evaluations are conducted on constrained RL benchmarks that primarily aim to maximize returns while keeping costs within predefined thresholds.
>
> Please note that our primary objective is to avoid unsafe policies without access to reward or cost information. Such an objective is quite important in safety critical environments such as health care and autonomous driving.
>
> To address reviewers concern of our results being conservative, we have conducted additional investigations and experiments, which are now included in Section C5 of the appendix (page 26 in the revised draft). In this section, we explore how to control the conservativeness of UNIQ by adjusting a scalar parameter in the Weighted Behavior Cloning (WBC) formulation. Our results demonstrate that this approach effectively balances safety and performance, allowing the policy to achieve higher rewards while remaining safer than other baselines.
>
> It is also worth noting that, since reward and cost information is unavailable (a common scenario in practical applications where experts or non-experts cannot explicitly specify reward or cost functions and instead provide demonstrations), achieving performance comparable to constrained RL methods with access to these signals is inherently challenging. This limitation underscores the unique difficulties of imitation learning in such settings and highlights the importance of developing methods tailored to these constraints.

---

> ### Author Response · Authors · 2024-11-24
>
> > As discussed in Section 1, the authors may argue that UNIQ is advantageous in certain scenarios — such as when expert demonstrations are entirely absent in the mixed dataset or when specific undesirable actions must be avoided by pretrained agents without compromising performance — the absence of experiments on these contexts weakens the argument for its applicability...
>
> We thank the reviewer for the suggestion. Experiments with real-world data, where undesirable demonstrations are available alongside good/expert data, would indeed be valuable. While such data and scenarios are often present in practice, we believe it is quite challenging at this stage to test our algorithm with such data.
>
> Beyond the inherent difficulties of accessing high-quality real-world datasets, policy evaluation in real-world settings is extremely challenging and costly. This is particularly true in critical domains like autonomous driving and healthcare, where incorrect decisions can have serious consequences, making such evaluations infeasible for researchers in academia.
>
> Moreover, UNIQ is the first algorithm in the imitation learning literature specifically designed to handle the "learning from undesirable demonstrations" scenario. In line with prior work, we have chosen to validate our approach on well-designed and widely used benchmarks. These benchmarks provide a controlled environment to rigorously test the utility of our method and demonstrate its advantages over existing approaches. We believe this is a necessary step before transitioning to more complex real-world evaluations.
>
> > Q1. In line 220, the manuscript assumes that “DMIX may contain a mix of random, undesired, and expert demonstrations.” However, datasets used in experiments comprised with undesired and expert demonstrations. Could you provide additional exprimental results with mixing (safe) random demonstrations to the dataset?
>
> Thank you for the suggestion. To address this, as well as the concern regarding publicly available datasets, we conducted additional experiments on the D4RL benchmark, where we incorporated random trajectories into the mixed dataset. The results, presented in Section C6 of the appendix (page 27 in the revised draft), demonstrate that UNIQ still outperforms other main baselines, showcasing its robustness and effectiveness even in the presence of noisy data.
>
> > Q2. Could you clarify what is meant by “the quality of the unlabeled dataset” in line 244? Does this refer to the proportion of preferred and non-preferred demonstrations...
>
> We refer to quality in terms of accumulated rewards and costs—in other words, the quality of the policy represented by the mixed dataset.
>
> SafeDICE's objective is to mimic a policy representing a mixture of unsafe and mixed demonstrations, making it highly dependent on the quality of the policy that represent the mixed dataset. For instance, if the mixed dataset consists entirely of undesirable demonstrations, SafeDICE will learn a policy that replicates this undesirable behavior, leading to undesired outcomes. In contrast, the primary objective of our UNIQ approach is to avoid undesirable behavior while utilizing mixed datasets to support sample efficiency, so is less dependent on the “quality” of the mixed dataset.
>
> -----------
>
> **We hope our revisions and our response  address the reviewers’ concerns and further clarify our contributions. If there are any additional questions or comments, we would be happy to address them.**

---

> > ### Comment · Reviewer_AdDM · 2024-11-27
> >
> > I appreciate the authors' efforts in addressing my concerns. However, I still have the following unresolved concerns:
> >
> > **1. Ambiguously Guided Safe Objective**
> >
> > As noted by other reviewers as well, it still remains challenging to agree that the proposed objective is practically useful.
> > The authors suggest that it could be beneficial when data is explicitly labeled as undesirable according to the learner's intention. However, such a scenario seems to require more data points compared to standard IL problems.
> > This is because, in many cases, the number of desirable actions per state is significantly smaller than the number of undesirable ones, especially with a deterministic expert.  For example, if $|A|=4$ (e.g. navigation task) there is only one optimal action, while the other three are undesirable.
> >
> > **2. Experimental Results on Mixed (Desirable+Random+Undesirable) Datasets**
> >
> > Thank you for the additional experiment involving random demonstrations added to the dataset (included in Section C.6).
> > However, as discussed by the authors, UNIQ demonstrates effectiveness only in the HalfCheetah task (one out of the two tasks tested). Consequently, it is difficult to draw a strong conclusion about the effectiveness of UNIQ in mixed datasets.
> >
> > For these reasons, I maintain my current rating for this work.

---

### Official Review · Reviewer_aLRo · 2024-11-09

**Soundness:** 4
**Presentation:** 4
**Contribution:** 4
**Rating:** 8
**Confidence:** 4

**Summary:**

This paper introduces a novel approach to learning from "undesirable" demonstrations. Instead of conventional imitation learning, which aims to imitate expert demonstrations, i.e. minimizing the distance with desired demonstrations, this method focuses on avoiding negative behaviors, i.e. maximizing the distance with undesired policy.

Based on MaxEnt RL framework and inverse Q-learning, the authors propose UNIQ, an offline inverse Q-learning algorithm designed to maximize the statistical distance between the target policy and undesirable policy. UNIQ utilizes both demonstrations of undesirable behavior and a broader pool of unlabeled data to develop a learning policy that effectively avoids unwanted actions.

**Strengths:**

Problem novelty - This paper tackles the distinct challenge of learning policies that actively avoid undesirable behaviors, rather than merely replicating expert behaviors. This approach is valuable in many applications where high-quality expert data might be scarce or unavailable.

Crisp and well-rounded introduction - Good summary of the problem space, the traditional approach, and the gap it's trying to address. Clearly explained frameworks that the algorithm is based on (inverse Q-learning).

Implementation details - The paper provides source code along with comprehensive details on hyper parameter settings and experimental configurations, improving both reproducibility and transparency.

**Weaknesses:**

Lack of validation using real-world data - While UNIQ is assessed on well-known benchmarks, evaluating it with real-world datasets from fields such as healthcare or autonomous driving could greatly enhance its value and demonstrate its applicability beyond simulated settings.

**Questions:**

Have you considered testing UNIQ in real-world domains? What specific challenges do you foresee in transferring the method from simulated benchmarks to real-world datasets?

---

> ### Author Response · Authors · 2024-11-24
> **We thank the reviewers for the feedback!**
>
> We thank the reviewer for the detailed comments. Please find our responses below.
>
> -----------
> >Lack of validation using real-world data - While UNIQ is assessed on well-known benchmarks, evaluating it with real-world datasets from fields such as healthcare or autonomous driving could greatly enhance its value and demonstrate its applicability beyond simulated settings.
>
> > Have you considered testing UNIQ in real-world domains? What specific challenges do you foresee in transferring the method from simulated benchmarks to real-world datasets?
>
> We understand the reviewer’s concern and truly get that it will enhance its value. However, there are two main reasons for focusing on well-studied benchmarks and not on real world datasets:
> - Policy evaluation with real world data is extremely challenging and costly (given the impact of wrong decisions), especially in autonomous driving and healthcare domains, that is not feasible for researchers in academia.
> - To that end, to make progress in imitation learning, researchers have developed well-rounded and extensive benchmarks to evaluate whether a new approach provides significant utility beyond existing approaches. We also employ the same methodology of evaluation.
> -----------
> **We hope our responses address your concerns. If there are any additional questions or comments, we would be happy to address them.**

---

### Author Response · Authors · 2024-11-24
**We thank the reviewers for their feedback!**

We sincerely thank the reviewers for their constructive and thoughtful feedback, which we have addressed to the best of our ability. We have provided a comprehensive set of new results to respond to the concerns raised. To address the reviewers’ concerns, we have made the following major updates to the paper:

- Added experiments and discussions on the conservativeness of UNIQ (Section C5 in the appendix, page 26 in the revised draft).

- Included additional experiments using the publicly available D4RL dataset (Section C6, page 27 in revised draft).

- Provided a discussion and some experimental results regarding the public DSRL dataset (section C7, page 27 in revised draft).



We hope these revisions address the reviewers’ critiques and provide greater clarity regarding our contributions. If there are any additional questions or comments, we would be happy to address them.

---

### Meta-Review · Area_Chair_tVGx · 2024-12-20

**Metareview:**

(a) Summary: This work tackles the challenge of offline learning in scenarios with undesirable demonstrations, aiming to train agents that avoid unsafe or undesired behaviors in environments where undesirable and mixed demonstrations are given. The paper presents UNIQ (UNdesirable demonstrations-driven Inverse Q-Learning), an approach built on an inverse Q-learning (IQ-Learn) framework.
(b) Strengths: The paper is generally well-written and easy to follow. The contributions seem to be interesting and reasonable.
(c) Weaknesses: The reviewers pointed out a few major concerns and issues. The novelty is incremental. Some important experimental comparisons are insufficient. The authors did not reply to address the remaining issues after the reviewers replied to the rebuttal.

**Additional Comments On Reviewer Discussion:**

Although the reviewers tried to address the reviewers' concerns during the rebuttal, there were still many issues unsolved. The authors did not provide further response.

---

### Decision · Program_Chairs · 2025-01-22

Reject